# Widespread fungal–bacterial competition for magnesium lowers bacterial susceptibility to polymyxin antibiotics

Yu-Ying Phoebe Hsieh[1]*, Wanting Sun[1], Janet M. Young[1], Robin Cheung[2], Deborah A. Hogan[3], Ajai A. Dandekar[2,4], Harmit S. Malik[1,5]*

1 Division of Basic Sciences, Fred Hutchinson Cancer Center, Seattle, Washington, United States of America, 2 Department of Microbiology, University of Washington, Seattle, Washington, United States of America, 3 Department of Microbiology and Immunology, Geisel School of Medicine at Dartmouth, Hanover, New Hampshire, United States of America, 4 Department of Medicine, University of Washington, Seattle, Washington, United States of America, 5 Howard Hughes Medical Institute, Fred Hutchinson Cancer Center, Seattle, Washington, United States of America

☯ These authors contributed equally to this work.

* yhsieh@fredhutch.org (Y-YPH); hsmalik@fredhutch.org (HSM)

**Data Availability Statement:** All relevant data are within the paper and its Supporting information files. Sequencing files are stored in the BioProject PRJNA1021673 of the NCBI repository. Analysis

## Abstract

Fungi and bacteria coexist in many polymicrobial communities, yet the molecular basis of their interactions remains poorly understood. Here, we show that the fungus *Candida albicans* sequesters essential magnesium ions from the bacterium *Pseudomonas aeruginosa*. To counteract fungal $Mg^{2+}$ sequestration, *P. aeruginosa* expresses the $Mg^{2+}$ transporter MgtA when $Mg^{2+}$ levels are low. Thus, loss of MgtA specifically impairs *P. aeruginosa* in co-culture with *C. albicans*, but fitness can be restored by supplementing $Mg^{2+}$. Using a panel of fungi and bacteria, we show that $Mg^{2+}$ sequestration is a general mechanism of fungal antagonism against gram-negative bacteria. $Mg^{2+}$ limitation enhances bacterial resistance to polymyxin antibiotics like colistin, which target gram-negative bacterial membranes. Indeed, experimental evolution reveals that *P. aeruginosa* evolves *C. albicans*-dependent colistin resistance via non-canonical means; antifungal treatment renders resistant bacteria colistin-sensitive. Our work suggests that fungal–bacterial competition could profoundly impact polymicrobial infection treatment with antibiotics of last resort.

## Introduction

Competition is a pivotal force shaping microbial life. To thrive in crowded microbial communities, microbes have evolved a wide range of strategies to compete for resources or suppress the fitness of their neighbors, for example, by using iron scavengers [1], contact-dependent inhibition [2], or the production of antibiotics [3–5]. Extensive studies on inter-bacterial competition have revealed profound impacts of inter-species competition on the diversity of bacterial strains, coevolution of bacterial species, and species composition within microbial communities [6]. Competition between fungi and bacteria has been relatively less well studied despite the fact that fungi and bacteria cohabit many polymicrobial environments, from soils

scripts are available in the GitHub repository (https://github.com/PhoebeHsieh-yuying/P.-aeruginosa_Tnseq_paper) and the Zenodo repository (DOI: 10.5281/zenodo.11404260). Strains and plasmids are available upon request.

**Funding:** This work was supported by a postdoctoral fellowship from the Cystic Fibrosis Foundation (HSIEH21F0 to Y-YPH) and grants from the National Institutes of Health (R01GM125714, R35GM152107 to AAD; R01AI127548 to DAH), from the Burroughs Wellcome Fund (1012253 to AAD), and from the Howard Hughes Medical Institute (Investigator award to HSM). The funders had no role in study design, data collection and analysis, decision to publish, or preparation of the manuscript.

**Competing interests:** HSM is a member of the PLOS Biology Editorial Board.

**Abbreviations:** BHI, brain heart infusion; CFU, colony-forming unit; FDR, false discovery rate; LPS, lipopolysaccharide; SCFM, synthetic cystic fibrosis medium; TSB, tryptic soy broth; UTR, untranslated region; WT, wild-type.

[7] to cheese [8] to host-associated niches, such as the human gut or polymicrobial infections [9].

Accumulated evidence suggests that fungi or bacteria use specific strategies to sense or combat each other. For instance, fungal–bacterial coexistence is known to drive the production of anti-bacterial [10] or anti-fungal [11] metabolites. Some bacteria produce toxins that can target certain fungal species [12,13]. A recent study based on a synthetic microbial community revealed that fungi can simultaneously suppress and promote bacterial fitness [14]. Fungal–bacterial competition has also been reported in some metagenomic studies [15]. However, fungal–bacterial competition is hard to dissect in complex microbial communities. As a result, the underlying molecular mechanisms of fungal–bacterial competition remain largely unknown. A greater understanding of fungal–bacterial competition could reveal novel insights that enhance our ability to predict the biological outcomes of fungal–bacterial competition and help devise strategies to control fungal–bacterial interactions within diverse microbial communities.

Here, we focus on the fungus *Candida albicans* and the bacterium *Pseudomonas aeruginosa*, 2 opportunistic microbes that are frequently found together in close contact in diverse sites including in biofilms [16,17], chronic wounds [18], and the airways of people afflicted with cystic fibrosis [19,20]. Previous studies have identified strategies that *C. albicans* and *P. aeruginosa* use to antagonize each other, including bacterial toxins that target fungal hyphae [17,21], interference with quorum-sensing regulation [22,23], and competition for limited resources like iron [24]. Most studies have focused on anti-fungal strategies imposed by bacteria, whereas fungal strategies for competing with bacteria remain less well-understood. Similarly, it is unclear whether or how fungal–bacterial competition influences the evolution of drug resistance. Identification of strategies used by fungi to antagonize bacteria, potential bacterial counterstrategies, and the consequences of such fungal–bacterial competition on antibiotic resistance could be leveraged to predict resistance development in polymicrobial infections and improve therapies to cure infectious diseases.

## Results

To identify strategies used by fungi to antagonize bacteria, we first investigated whether co-culture with *C. albicans* (strain SC5314) affects the fitness of *P. aeruginosa* (strain PAO1) in brain heart infusion (BHI) broth, a medium commonly used for isolating microbes from clinical samples or studying human microbial pathogens in vitro. We found that co-culture with PAO1 did not impair *C. albicans* fitness, whereas co-culture with *C. albicans* impaired PAO1 fitness in BHI media 10- to 100-fold relative to bacteria-only conditions (monoculture) (S1A Fig). Previous studies have investigated *P. aeruginosa* transcriptomes in a variety of media but not in BHI broth [25]. Similarly, *P. aeruginosa* transcriptomes in co-culture with *C. albicans* have not been investigated in BHI broth [26]. Therefore, to understand the physiological basis for *P. aeruginosa* impairment in co-culture with *C. albicans* in BHI media, we performed an RNA-seq analysis on *P. aeruginosa* following 8 h of co-culture with *C. albicans*, relative to monoculture. We chose this time point for the transcriptome analysis as bacterial fitness was nearly identical between the 2 conditions at this time (S2 Fig). Our analysis revealed that 145 *P. aeruginosa* genes were up-regulated by at least 4-fold in co-culture relative to monoculture conditions, including those related to TonB-dependent substrate transport, siderophore synthesis, and RNA polymerase sigma factor 70 (S3A Fig and S1 Table). We also found that 134 genes, including those for Type VI secretion system, co-factor biosynthesis, and energy generation, were down-regulated by at least 4-fold in co-culture (S3B Fig and S1 Table). These

changes in gene expression suggest that *P. aeruginosa* cells prioritized nutrient uptake while minimizing energy expenditure in co-culture with *C. albicans*.

We hypothesized that *P. aeruginosa* might rely on fungal-defense genes to protect itself during co-culture with *C. albicans*; loss of such fungal-defense genes would further impair *P. aeruginosa* fitness under co-culture conditions [27]. To identify genes important for bacterial fitness in the presence of fungi, we used a transposon-insertion sequencing (Tn-seq) approach [28–30] to conduct a genome-wide fitness screen in bacteria. We cultured a pool of $10^5$ unique *P. aeruginosa* transposon-insertion (Tn) mutants [28] for 10 generations, either in monoculture or in co-culture with *C. albicans*. We then quantified the fitness of each Tn mutant by comparing the number of reads of each transposon mutant in both conditions (Fig 1A). Our findings revealed that Tn insertions in 8 *P. aeruginosa* genes significantly reduced bacterial fitness in co-culture relative to monoculture conditions (Fig 1B and 1C). Three adjacent genes— *PA4824*, *PA4825*, and *PA4826*—showed the most significant fitness loss in co-culture (Fig 1B). These 3 genes were also the only overlap between our Tn-seq and RNA-seq analyses, with *PA4824* and *PA4825* showing a 32-fold increase in expression in co-culture (S3C Fig). Of these 3 genes, only *PA4825* has been functionally characterized; it encodes a magnesium transporter known as MgtA [31]. In addition to the 8 genes in which Tn insertions decreased fitness (putative fungal-defense genes), our Tn-seq analyses identified 18 genes whose loss enhanced bacterial fitness in co-culture (S2 Table), suggesting these genes either become "dispensable" or incur a fitness cost in co-culture conditions.

## Magnesium is an axis of nutritional competition between *P. aeruginosa* and *C. albicans*

We focused on the contributions of the *PA4824*, *PA4825 (mgtA)*, and *PA4826* genes during co-culture. We engineered gene deletion mutants and measured the fitness of these mutants relative to a fluorescently labeled wild-type (WT) strain in either monoculture or co-culture with *C. albicans* using competitive fitness assays, which resemble the pooled Tn-seq bulk selection conditions (S4A Fig). Our experiments showed that the relative fitness of the Δ*PA4824* or Δ*PA4825* (Δ*mgtA*) mutants, but not Δ*PA4826*, was approximately 10% lower than the WT in co-culture (S4B Fig). In contrast, neither of these mutants had any measurable fitness defect, relative to WT PAO1, in monoculture (S4B Fig). Co-culture colony formation assays also confirmed our findings; deletion of either Δ*PA4824* or Δ*mgtA* led to a 10-fold reduction in colony-forming units (CFUs) of *P. aeruginosa* in co-culture relative to the CFUs of WT in co-culture (Fig 1D). CFUs of Δ*PA4824* or Δ*mgtA* in co-culture were restored by expressing a WT copy of *PA4824* or *mgtA* in trans (S4C and S4D Fig). Moreover, a double deletion Δ*PA4824* Δ*mgtA* mutant was even more significantly impaired than single deletion mutants (Fig 1D). These results suggest that the protein products encoded by *PA4824* and *mgtA* are independently required for optimal *P. aeruginosa* fitness under co-culture conditions, but *PA4826* is not. We infer that the identification of *PA4826* in our Tn-seq analyses may be due to the effects of insertional mutants in *PA4826* on the expression of adjacent *PA4824* and *PA4825 (mgtA)* genes.

Our finding that *P. aeruginosa PA4825 (mgtA)* is required for full bacterial fitness in co-culture led us to hypothesize that *C. albicans* depletes the vital $Mg^{2+}$ cation in co-culture. Under $Mg^{2+}$ depletion conditions, *mgtA* might become essential for fitness since its loss would result in lower intracellular $Mg^{2+}$ levels. To test this hypothesis, we employed a $Mg^{2+}$ genetic sensor [31] from *Salmonella enterica* serovar Typhimurium. Previous studies showed that *S*. Typhimurium *mgtA* expression levels are controlled by its 5′ untranslated region (5′UTR), which acts as a ribo-switch, adopting different stem-loop structures depending on the $Mg^{2+}$ levels and regulating the transcription of *mgtA* [31]. We showed that this *S*. Typhimurium $Mg^{2+}$

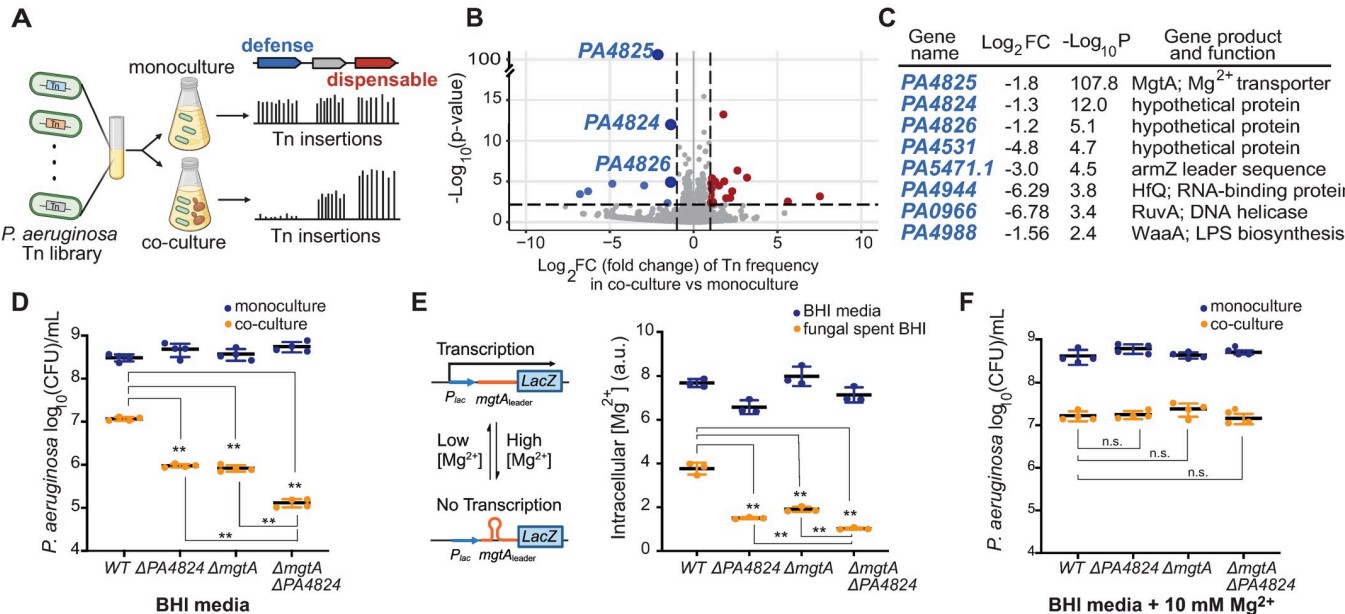

**Fig 1. *P. aeruginosa* fitness is suppressed by *C. albicans*-mediated Mg²⁺ sequestration.** (A) A pool of *P. aeruginosa* Tn mutants was grown in BHI media only or in co-culture with *C. albicans* in BHI. Quantification of Tn mutant frequencies (see Methods) revealed genes whose loss either impaired *P. aeruginosa* fitness (blue; fungal-defense genes) or conferred a fitness advantage (red; dispensable genes) in co-culture with *C. albicans*. (B) Tn-seq volcano plot (see Methods) shows *P. aeruginosa* genes important for defense or dispensable in co-culture with *C. albicans*. X-axis indicates fold change, while Y-axis indicates $p$-value after correcting for multiple testing. We used $\log_2$ fold change > 2 and adjusted $p$-value < 0.1 as statistical cutoffs. (C) Candidate defense genes identified by Tn-seq. A full list of these genes is available in S2 Table. (D) Fitness of *P. aeruginosa* single deletion Δ*PA4824* or Δ*mgtA* mutants was impaired relative to WT *P. aeruginosa* in co-culture with *C. albicans* (orange) but not monoculture in BHI media (blue). The fitness of double deletion Δ*PA4824* Δ*mgtA* mutant was even further impaired than single deletion mutants. CFUs of *P. aeruginosa* were measured by serially diluting cultures on LB+Nystatin. (E) (left panel) Intracellular Mg²⁺ in *P. aeruginosa* is measured using an RNA sensor. When the intracellular Mg²⁺ level is sufficient, *mgtA* 5′UTR forms a secondary structure that blocks downstream transcription. In limiting Mg²⁺, this structure is resolved, and transcription of a β-galactosidase reporter is restored (right panel). Intracellular Mg²⁺ levels, measured in β-galactosidase units, of single deletion mutants lacking *PA4824* or *mgtA* were lower than the WT stain and even lower in a double deletion Δ*PA4824* Δ*mgtA* mutant in *C. albicans*-spent BHI media (orange), but not BHI media alone (blue). (F) Co-culture-specific fitness of the *P. aeruginosa* single or double deletion mutants relative to WT strains was restored by Mg²⁺ supplementation (10 mM) in BHI in co-culture (orange). Mean ± std of 3 biological replicates is shown in panels D–F. (** $p$ < 0.01 Dunnett's one-way ANOVA test used; n.s. indicates not significant). The data underlying Fig 1D–1F can be found in S6 Data. Fig 1A and 1E are created with Biorender.com. BHI, brain heart infusion; CFU, colony-forming unit; WT, wild-type.

sensor can accurately detect changes in intracellular *P. aeruginosa* Mg²⁺ levels (S5 Fig). Using this sensor, we found that the intracellular Mg²⁺ levels in the WT *P. aeruginosa* are reduced by 2-fold in *C. albicans*-spent media (i.e., derived from the filtrate of *C. albicans* cultures to simulate Mg²⁺ depletion) compared to in fresh BHI media (Fig 1E). Loss of *mgtA* further reduced bacterial intracellular Mg²⁺ levels in the fungal spent media, but not in monoculture conditions (Fig 1E). Surprisingly, loss of *PA4824* also reduced intracellular Mg²⁺ levels in fungal spent media compared to the WT strain, with a double deletion Δ*PA4824* Δ*mgtA* mutant further reducing intracellular Mg²⁺ levels, compared to the single gene deletion mutants (Fig 1E). These experiments confirm fungal-imposed Mg²⁺ sequestration and demonstrate that *mgtA* and *PA4824* are independently required for both maintaining intracellular *P. aeruginosa* Mg²⁺ levels as well as fitness of *P. aeruginosa* cells in co-culture with *C. albicans*.

Our findings suggest that *PA4824* may play a crucial role in Mg²⁺ uptake. Although its function remains uncharacterized, Alphafold structural predictions [32] revealed that *PA4824* likely encodes a transmembrane protein with a distinctive β-barrel structure commonly associated with proteins involved in ion transport or nutrient uptake (S6A Fig). The core of PA4824 is composed of hydrophilic and negative-charged residues (S6B and S6C Fig), suggesting it might transport positive-charged molecules. Based on our co-culture RNA-seq and Tn-seq

experiments, results from the $Mg^{2+}$ genetic sensor assay, and the Alphafold prediction of PA4824 protein structure, we speculate that PA4824 potentially acts as a novel $Mg^{2+}$ transporter.

In addition to MgtA (and potentially PA4824), *P. aeruginosa* uses 2 other constitutively expressed $Mg^{2+}$ transporters, CorA and MgtE, to obtain $Mg^{2+}$ from the environments [33]. However, neither CorA nor MgtE was implicated in competition with *C. albicans* by our Tn-seq or RNA-seq analyses (S7A Fig). We also found that *corA* or *mgtE* loss-of-function mutants did not alter *P. aeruginosa* fitness in co-culture conditions (S7B Fig), nor did they significantly reduce intracellular $Mg^{2+}$ levels in *C. albicans*-spent BHI compared to Δ*mgtA* mutant (S7C Fig). Our experiments reveal that MgtA is the primary $Mg^{2+}$ transporter induced in co-culture, crucial for bacterial fitness and $Mg^{2+}$ uptake in co-culture. Thus, we conclude that that MgtA is a key bacterial $Mg^{2+}$ transporter required to overcome fungal sequestration of $Mg^{2+}$.

If *C. albicans* sequesters $Mg^{2+}$ from *P. aeruginosa*, we hypothesized that supplementing cultures with $Mg^{2+}$ might rescue the fitness defects of the Δ*mgtA* and Δ*PA4824* mutants. Indeed, supplemental $Mg^{2+}$ (10 mM) was sufficient to restore the fitness of Δ*mgtA*, Δ*PA4824*, and Δ*PA4824* Δ*mgtA* double deletion mutants to WT levels in co-culture conditions (Fig 1F). To investigate whether $Mg^{2+}$ could explain fitness effects of other candidate genes revealed by the Tn-seq analyses, we repeated our Tn-seq experiments in monoculture versus co-culture conditions except that we added 10 mM $Mg^{2+}$ to BHI media. To our surprise, we found that $Mg^{2+}$ supplementation ameliorated the fitness effects of most of the candidate genes identified in our initial Tn-seq analyses (S8A Fig), including all 8 fungal-defense genes and 17 of 19 dispensable genes crucial to fundamental cellular processes, including cell division, pyrimidine synthesis, and energy production—enzymes in essential processes that rely on $Mg^{2+}$. Despite their importance to bacterial physiology, we hypothesize that these otherwise critical processes might become too costly to maintain in low $Mg^{2+}$ co-culture conditions. Only 2 genes (*PA4029* and *PA5484)* remained marginally essential in co-culture, even under supplemented $Mg^{2+}$ conditions (S8A Fig). Notably, $Mg^{2+}$ supplementation only affected the expression of *PA4824-PA4826*, but not other candidate genes (S8B Fig), suggesting that $Mg^{2+}$ levels affect their functions posttranscriptionally. Thus, our finding underscores the influential role of $Mg^{2+}$ competition in mediating the fitness effects of most candidate genes in co-culture conditions. We note, however, that $Mg^{2+}$ supplementation did not fully restore the fitness of *P. aeruginosa* WT cells in co-culture to monoculture fitness levels (Fig 1F), consistent with prior observations of additional, $Mg^{2+}$-independent, axes of antagonism between bacteria and fungi [26,34,35].

Nutritional competition for important metal ions, such as iron, zinc, and manganese, is known for competition among microbes and between microbes and vertebrate hosts [36–38]. However, competition for the vital $Mg^{2+}$ cation is a novel finding. We reasoned that our finding might reflect different $Mg^{2+}$ levels in various media conditions. Indeed, we found that MgtA is required for *P. aeruginosa* fitness in co-culture with *C. albicans* in BHI and tryptic soy broth (TSB) media (S9A Fig), but not in yeast extract-peptone-dextrose (YPD) or synthetic cystic fibrosis media (SCFM) [39] (S9B and S9C Fig). $Mg^{2+}$ levels in BHI and TSB were lower (100 to 140 μm) than in YPD or SCFM (240 to 390 μm) (S9D Fig). As a result, fungal filtrates derived from BHI and TSB exhibited the lowest levels of $Mg^{2+}$ (S9D Fig), leading to a significant reduction of intracellular $Mg^{2+}$ in *P. aeruginosa* (S9E Fig). Additionally, by titrating $Mg^{2+}$ levels in co-culture, we found that supplementation of BHI with greater than 300 μm $Mg^{2+}$ was sufficient to relieve the fitness defect of the *P. aeruginosa* Δ*mgtA* mutant (S9F Fig). Thus, we conclude that BHI media represents a low $Mg^{2+}$ condition that reveals a novel mode of nutritional competition between fungi and bacteria.

MgtA was previously described as a $Mg^{2+}$-specific transporter [31]. To rule out the possibility that the fitness impairment of Δ*mgtA* mutant in co-culture could be caused by the

reduction of other cations in BHI, we performed a series of supplementation experiments with alternate cations. We found that $Mg^{2+}$ levels specifically determine *P. aeruginosa* viability in co-culture with *C. albicans*; supplementation of zinc ($Zn^{2+}$) or copper ($Cu^{2+}$) ions did not restore the viability of $\Delta mgtA$ mutant in co-culture to the monoculture levels (S9G Fig). The addition of molybdenum ($Mo^{6+}$) or manganese ($Mn^{2+}$) ions drastically decreased *P. aeruginosa* viability even in monoculture (S9G Fig), which rules out the possibility that either of these 2 ions was limited in BHI. Finally, we found that only a very high concentration of $Ca^{2+}$ (10 mM) rescued the fitness defect of the $\Delta mgtA$ mutant in co-culture (S9H Fig), orders of magnitude higher than the concentration of $Mg^{2+}$ required to rescue $\Delta mgtA$ fitness in co-culture (S9F Fig). These experiments further confirm that the fitness of $\Delta mgtA$ in co-culture conditions is primarily determined by $Mg^{2+}$ sequestration.

## Widespread $Mg^{2+}$ competition between fungi and gram-negative bacteria

We next assessed whether $Mg^{2+}$ competition is a general means of antagonism by fungi against bacteria. *Escherichia coli* and *S.* Typhimurium are gram-negative bacteria that coexist with *C. albicans* in the human gut [40]. *E. coli* encodes a single *mgtA* ortholog (like *P. aeruginosa*), whereas *S.* Typhimurium encodes 2 paralogs: *mgtA* and *mgtB*. We found that either a single deletion $\Delta mgtA$ in *E. coli* or a double deletion $\Delta mgtA$ $\Delta mgtB$ in *S.* Typhimurium severely reduced bacterial fitness in co-culture with *C. albicans*, and such fitness reduction was restored via $Mg^{2+}$ supplementation (Fig 2A and 2B).

Next, we expanded our analyses to 3 additional fungal species that coexist with *P. aeruginosa* in polymicrobial infections [41]: *Candida tropicalis*, *Candida parapsilosis*, *Candida glabrata*, as well as a non-pathogenic species, *Saccharomyces cerevisiae*. In each case, we found that the *P. aeruginosa* $\Delta mgtA$ mutant had reduced fitness relative to WT cells in fungal co-culture (Fig 2C), which could be restored by $Mg^{2+}$ supplementation (Fig 2D).

This mode of competition might be highly specific between fungi and diverse gram-negative γ-proteobacteria we have tested; it does not manifest in inter-bacterial competition. For example, lack of *mgtA* did not lower *P. aeruginosa* fitness in co-culture with bacteria that commonly co-exist in CF airways, such as gram-positive *Staphylococcus aureus* (Fig 2E) or gram-negative *Burkholderia multivorans* (Fig 2F). Similarly, *P. aeruginosa* $\Delta mgtA$ mutant had similar fitness as WT cells in co-culture with other gut-colonizing gram-negative like *E. coli* or *S.* Typhimurium (Fig 2F). Despite BHI representing a low-$Mg^{2+}$ condition, our results showed that different bacterial species do not necessarily compete for $Mg^{2+}$ during co-culture, suggesting fungi may have a higher demand or propensity for $Mg^{2+}$ uptake and sequester $Mg^{2+}$ more effectively than gram-negative bacteria. Overall, our findings confirm that competition for $Mg^{2+}$ is a widespread mode of fungal antagonism against gram-negative bacteria and that MgtA plays an important role in counteracting fungal-mediated $Mg^{2+}$ competition. Whether fungi can suppress gram-positive bacteria through the same mechanism of $Mg^{2+}$ competition remains an open question.

How do fungi impose competition for $Mg^{2+}$? We investigated whether the loss of *S. cerevisiae* genes in $Mg^{2+}$ uptake or sequestration restores the fitness of *P. aeruginosa* $\Delta mgtA$ mutant. *S. cerevisiae* cells store 80% of cellular $Mg^{2+}$ in their vacuole [42]. To investigate whether $Mg^{2+}$ sequestration in the fungal vacuole is involved, we generated a deletion mutant in Mnr2 [43], previously proposed to be a vacuolar $Mg^{2+}$ transporter in *S. cerevisiae*. We found that the fitness of *P. aeruginosa* $\Delta mgtA$ mutant was restored to WT levels in co-culture with the *mnr2Δ* mutant (Fig 2G). Yet, *S. cerevisiae* *mnr2Δ* mutant had similar fitness as WT *S. cerevisiae* both in media only as well as in co-culture (S10 Fig), suggesting that lack of Mnr2 does not cause a strong growth defect. In contrast, *P. aeruginosa* $\Delta mgtA$ fitness was not restored by competition

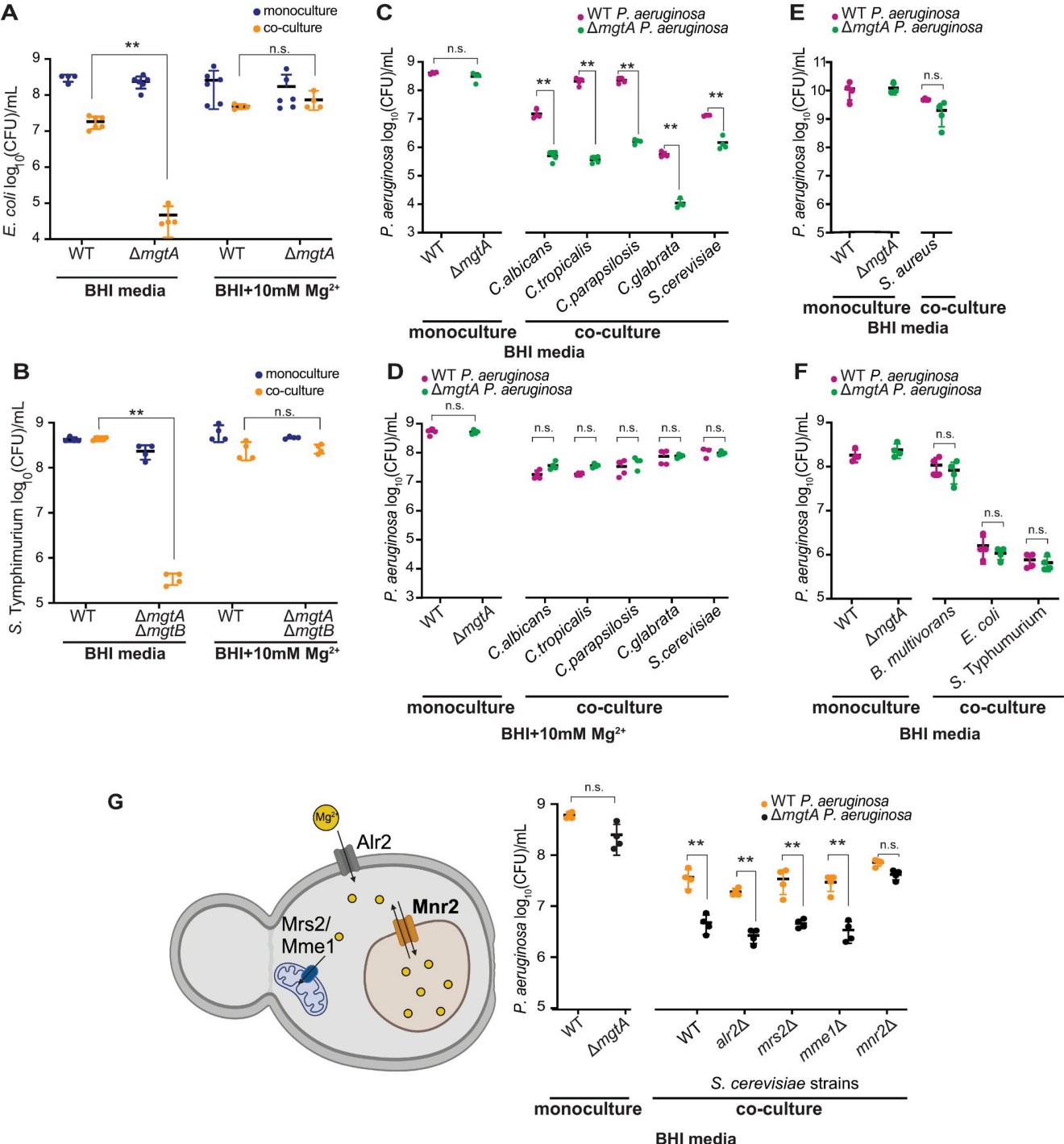

**Fig 2. Mg$^{2+}$ competition is widespread between multiple fungal and gram-negative bacterial species. (A)** The fitness of the *E. coli* Δ*mgtA* mutant was impaired relative to the WT strain in co-culture with *C. albicans* (orange) but not in monoculture (blue) in BHI media. The fitness of the Δ*mgtA* strain was restored in BHI media supplemented with 10 mM Mg$^{2+}$. **(B)** The fitness of the double deletion Δ*mgtA* Δ*mgtB* mutant *S.* Typhimurium strain was impaired relative to WT in co-culture with *C. albicans* (orange) but not in monoculture in BHI media (blue). **(C)** The fitness of the Δ*mgtA* mutant *P. aeruginosa* strain (green) was impaired relative to WT (magenta) in BHI in co-culture with fungal species: *C. albicans*, *C. tropicalis*, *C. parapsilosis*, *C. glabrata*, and *S. cerevisiae*. **(D)** Mg$^{2+}$ supplementation restored the fitness of the Δ*mgtA* mutant *P. aeruginosa* strain (green) in co-culture with fungal species in Fig 2C. **(E)** The fitness of the *P. aeruginosa* Δ*mgtA* mutant (green) does not significantly differ from WT (magenta) in co-culture with gram-positive *S. aureus*. **(F)** The fitness of the *P. aeruginosa* Δ*mgtA* mutant (green) was similar to WT (magenta) in co-culture with gram-negative bacteria, such as *B. multivorans*, *E. coli*, or *S.* Typhimurium. **(G)** The functions and locations of proposed Mg$^{2+}$ transporters in *S. cerevisiae* are illustrated with Biorender.com (top panel). Alr2 localizes to the plasma membrane, Mrs2 and Mme1 to the mitochondria, and Mnr2 to the vacuole. The fitness of the *P. aeruginosa* Δ*mgtA* mutant (black) is not

significantly different from the WT strain (orange) in monoculture. However, the relative fitness of *P. aeruginosa* Δ*mgtA* was impaired in co-culture with either WT *S. cerevisiae* or either of 3 *S. cerevisiae* mutant strains: Δ*alr2*, Δ*mrs2*, or Δ*mme1*. In contrast, the relative fitness of *P. aeruginosa* Δ*mgtA* was restored in co-culture with *S. cerevisiae* Δ*mnr2* strain. Mean ± std of 4 biological replicates is shown in panels A–G. (** $p < 0.01$ unpaired two-tailed Student's *t* test used; n.s. indicates not significant). The data underlying this Fig 2A–2G can be found in S11 Data. BHI, brain heart infusion; WT, wild-type.

with *S. cerevisiae* mutants lacking Alr2, a cell-wall-associated $Mg^{2+}$ transporter used to acquire environmental $Mg^{2+}$ into the cytosol [44], or those lacking Mrs2/Mme1, mitochondria-specific $Mg^{2+}$ transporters (Fig 2G) [45]. Thus, fungal vacuolar $Mg^{2+}$ sequestration might broadly mediate $Mg^{2+}$ antagonism against gram-negative bacteria.

## Fungal $Mg^{2+}$ sequestration protects bacteria from polymyxin antibiotics

$Mg^{2+}$ is required for many fundamental cellular processes by neutralizing negatively charged biomolecules and functioning as an enzyme cofactor [33]. $Mg^{2+}$ is also important for stabilizing the outer membrane structure of gram-negative bacteria by binding to negatively charged lipopolysaccharides (LPS) on the bacterial cell membrane [46]. These $Mg^{2+}$-stabilized LPS moieties can be targeted by polymyxins [47], last-resort antibiotics for treating multidrug-resistant bacterial infections [48,49] (Fig 3A). In $Mg^{2+}$-limited conditions, bacteria activate dual two-component signaling pathways, PmrAB and PhoPQ, modifying lipid A of LPS with L-Ara4N (4-amino-4-deoxy-L-arabinose) and PEtN (phosphoethanolamine), leading to resistance to polymyxins [50–53] (Fig 3A). Thus, fungal-mediated $Mg^{2+}$ depletion might confer polymyxin resistance in bacteria.

To test this possibility, we grew *P. aeruginosa* in fresh BHI media or *C. albicans*-spent BHI media, which mimics $Mg^{2+}$ depletion by fungi, and examined bacterial survival when treated with colistin (also referred to as polymyxin E) (Fig 3B). Consistent with our hypothesis, we found that cells grown in fresh media showed no survival after colistin treatment, whereas 20% of cells grown in fungal spent media survived colistin treatment (Figs 3C and S11A). Supplementation with extra $Mg^{2+}$ in *C. albicans*-spent media restored the sensitivity of *P. aeruginosa* cells to colistin, to levels resembling sensitivity in BHI fresh media (Figs 3C and S11A). Since low $Mg^{2+}$ conditions activate the PmrAB and PhoPQ signaling pathways required to modify LPS and confer colistin resistance, we tested whether *pmrB or phoQ* loss-of-function transposon mutants [54] would abolish the observed colistin resistance in fungal spent media; we found that this was the case (Fig 3C).

We found that fungi-dependent increased colistin survival is not restricted to *P. aeruginosa* or colistin. For example, we found that $Mg^{2+}$ depletion by *C. albicans* also increased the survival of *S.* Typhimurium to colistin, a phenotype that is also suppressed by extra $Mg^{2+}$ (Figs 3D and S11B). Furthermore, $Mg^{2+}$ depletion increased *P. aeruginosa* survival to another polymyxin antibiotic, polymyxin B (S12 Fig). Thus, fungal sequestration of $Mg^{2+}$ provides a general means of conferring polymyxin resistance on gram-negative bacteria.

Since fungal spent media cannot fully recapitulate fungal presence in co-culture conditions, we tested whether fungal co-culture also conferred increased colistin survival (Fig 3E). We found that co-culture with *C. albicans* was sufficient to protect *P. aeruginosa* and *S.* Typhimurium from colistin across a range of concentrations (1.5 μg/ml to 12.5 μg/ml; the clinical MIC of *P. aeruginosa* to colistin is 2 μg/ml), in contrast to monoculture conditions at both 30 ˚C (Figs 3F, 3G, S13A and S13B) and 37 ˚C (S14A and S14B Fig). Additionally, the protection from antibiotics by fungi in co-culture applies to different *P. aeruginosa* strains. We found that co-culture with *C. albicans* also protects several colistin-sensitive *P. aeruginosa* isolates from cystic fibrosis patients from colistin (S15 Fig). Thus, *C. albicans* can protect gram-negative bacteria from colistin across a range of conditions.

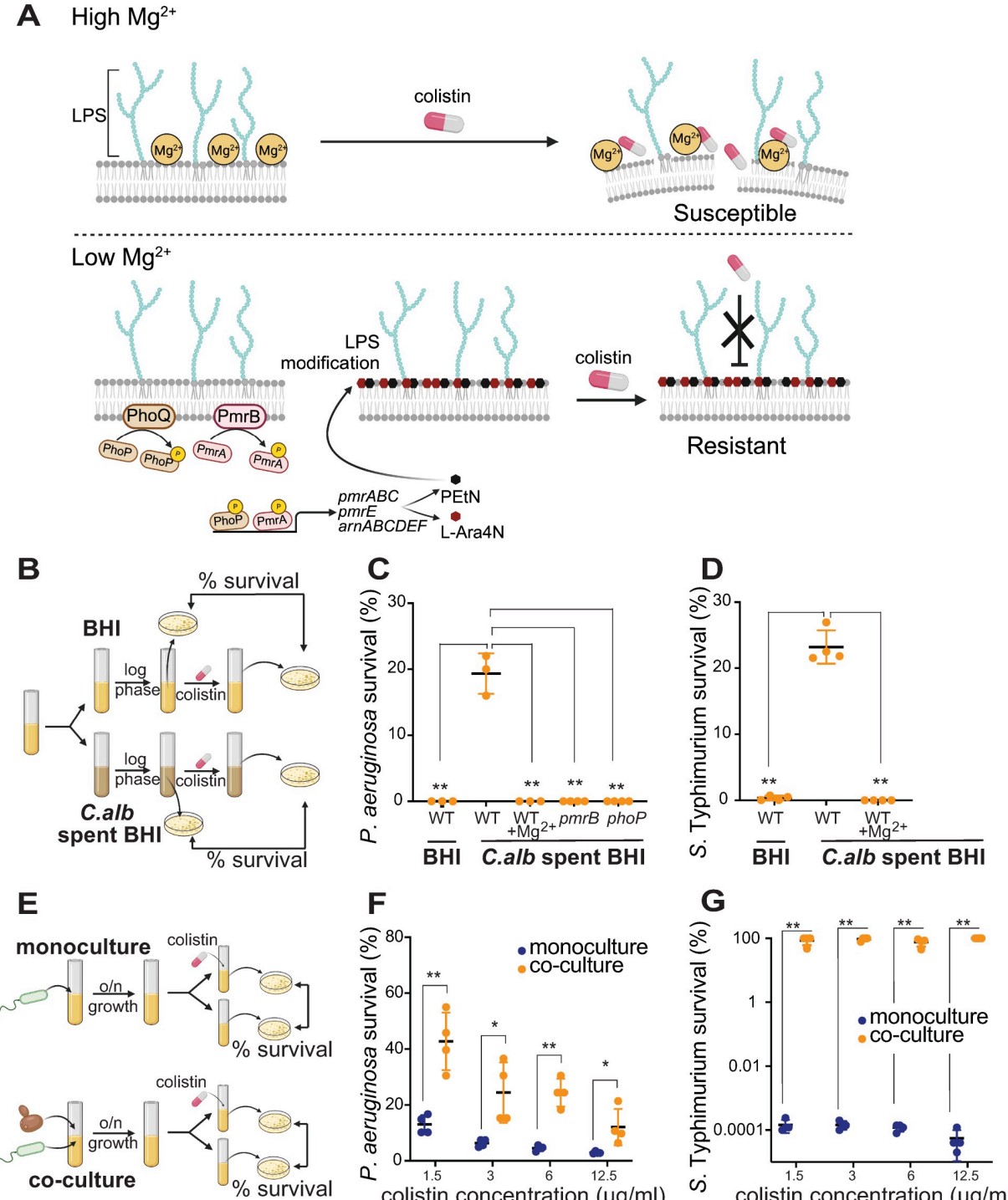

**Fig 3. *C. albicans*-mediated Mg²⁺ sequestration protects gram-negative bacteria from colistin.** **(A)** $Mg^{2+}$ binds to negatively charged lipid A of LPS on the outer membrane of gram-negative bacteria. In high $Mg^{2+}$ conditions, colistin (and other polymyxin antibiotics) targets $Mg^{2+}$-bound LPS to disrupt cell membranes of gram-negative bacteria, leading to bacterial cell death. Low $Mg^{2+}$ conditions activate two-component signaling pathways, PmrAB and PhoPQ, that modify LPS (PEtN and L-Ara4N), making bacteria resistant to colistin. **(B)** Schematic of the colistin survival assay in *C. albicans*-spent media. *P. aeruginosa* strains were grown to stationary phase in BHI, *C. albicans*-spent BHI, or *C. albicans*-spent BHI supplemented with 10 mM $Mg^{2+}$, and then passaged until exponential phase, when they were treated with 25 μg/ml colistin. Bacterial survival is shown as the percentage of viable cells after the colistin treatment. **(C)** WT *P. aeruginosa* strains were highly susceptible to colistin treatment in BHI media. However, they could survive colistin treatment in *C. albicans*-spent BHI media but not in spent BHI media supplemented with 10 mM $Mg^{2+}$. In contrast to WT strains, *pmrB* or *phoP* Tn-mutants of *P. aeruginosa* (which cannot modify LPS) remained highly susceptible to colistin

even in *C. albicans*-spent BHI media. **(D)** WT *S.* Typhimurium was highly susceptible to 3 μg/ml colistin in either BHI media or *C. albicans*-spent media supplemented with 10 mM Mg$^{2+}$ but became more resistant to colistin in fungal spent media alone. **(E)** Schematic of the colistin survival assay in co-culture. The *P. aeruginosa* WT strain was cultured only in BHI or co-culture with *C. albicans* for 18 h. Monocultures or co-cultures were then treated with colistin. Bacterial survival is calculated as the percentage of viable cells after the colistin treatment, just as in (B). **(F)** *P. aeruginosa* co-cultures with *C. albicans* (orange) were more resistant than monocultures (blue) over a series of colistin concentrations, from 1.5 μg/ml colistin (clinical MIC) to 12.5 μg/ml colistin (8× clinical MIC). **(G)** Like in (F), co-cultures of *S.* Typhimurium with *C. albicans* (orange) were significantly more resistant than monocultures of *S.* Typhimurium alone over a range of colistin concentrations. Mean ± std of 3 biological replicates is shown in panels C–D and F–G. (**$p < 0.01$, Dunnett's one-way ANOVA test and unpaired two-tailed Student's *t* test were used in panel C–D and F–G separately). The data underlying this Fig 3C–3D and 3F–3G can be found in S13 Data. Fig 3A, 3B, and 3E are created with Biorender.com. BHI, brain heart infusion; LPS, lipopolysaccharide; WT, wild-type.

## Fungal co-culture fundamentally alters the evolution of bacterial colistin resistance

Understanding how microbial interactions drive the evolution of antibiotic resistance is important for biological and biomedical reasons; increased antibiotic resistance is often observed in polymicrobial communities [55]. We hypothesized that long-term coexistence with *C. albicans* may fundamentally alter the onset and evolution of enhanced colistin survival in *P. aeruginosa*. To test this possibility, we passaged 8 replicate WT populations of *P. aeruginosa* PAO1 for 90 days, either in monoculture or in co-culture with *C. albicans*, with each population exposed to gradually increasing concentrations of colistin (from 1.5 μg/ml to 192 μg/ml) (Fig 4A). As controls, we also passaged mono- and co-culture populations in the absence of colistin (Fig S16). After evolution, both monoculture- and co-culture-evolved populations gained increased survival to 192 μg/ml colistin (Fig 4B). Surprisingly, we found that enhanced colistin survival in *P. aeruginosa* co-culture populations was dependent on *C. albicans*; colistin survival was almost completely lost upon removal of *C. albicans* using antifungal treatment or the addition of Mg$^{2+}$ (Figs 4C and S17). Our findings indicate that co-culture-evolved populations acquire increased colistin survival due to fungal protection induced by Mg$^{2+}$ sequestration but continue to depend on the fungus for increased protection.

Based on our findings, we suspected that *P. aeruginosa* populations evolved in co-culture with fungi likely enhance colistin survival via fundamentally different mutational mechanisms than monoculture-evolved populations. To test this hypothesis, we performed genome sequencing of all experimentally evolved populations. We found that all colistin-treated populations grown in monoculture became hypermutators (Fig 4D) by acquiring at least 1 mutation in genes involved in DNA replication or repair (*uvrB*, *recQ*, and *mutS*) (Fig 4E and S3 Table). Hypermutation is a hallmark of antibiotic-resistance acquisition that has been observed in several previous experimental evolution studies [56]. As a result of hypermutation, each monoculture population acquired hundreds of fixed mutations (Fig 4D), including mutations previously shown to confer colistin resistance, in genes such as *pmrB* [57], *ptsP* [58,59], *pqsR* [58], *colS* [60,61], porins [58], and proteins related to bacterial motility [62] (Fig 4E and S3 Table). In contrast, none of the co-culture-evolved populations became hypermutators; each only harbored 6 to 8 fixed mutations (Fig 4D), primarily in other targets for colistin resistance, such as genes in LPS biosynthesis [63] and an outer-membrane protein *oprH* [52] (Figs 4E and S18 and S3 Table). Intriguingly, we found that *PA4824*, another co-culture-specific defense gene identified in our Tn-seq experiments, was also mutated in all 8 replicates of co-culture-evolved populations (Figs 4E and S14 and S3 Table). We did not find fixed mutations in any of these genes in co-culture populations that were not exposed to colistin (S4 Table). Thus, the dual stressors of colistin and fungal co-culture specifically select for distinct classes of adaptive mutations, fundamentally altering the mode of bacterial colistin survival.

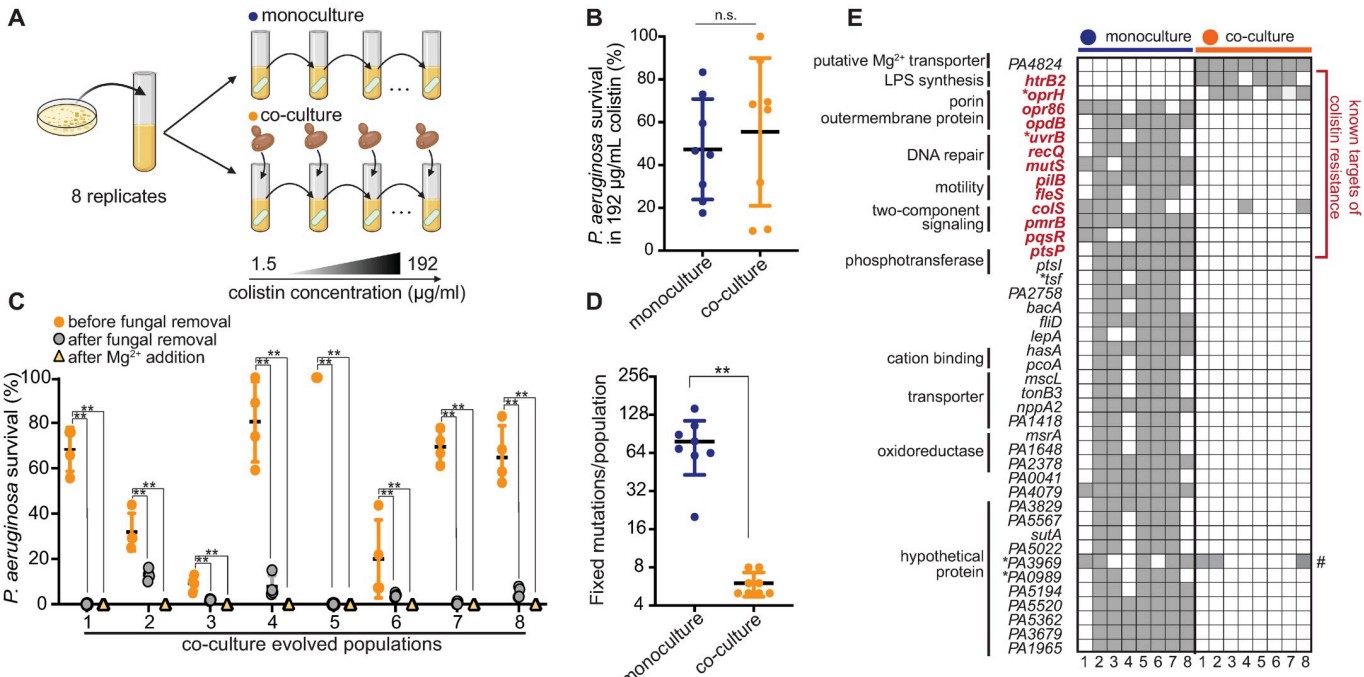

**Fig 4. Long-term coexistence with *C. albicans* alters the evolution of *P. aeruginosa* colistin resistance. (A)** Schematic of the evolution experiment in co-culture, created with Biorender.com. Eight independent WT *P. aeruginosa* populations were passaged daily in BHI with colistin or in BHI with both *C. albicans* and colistin. Colistin levels were gradually increased from 1.5 μg/ml to 192 μg/ml ($128×$ clinical MIC) over 90 days (see Methods). **(B)** After evolution, 8 monoculture replicate populations (blue) were not significantly different from 8 co-culture populations (orange) for resistance to 192 μg/ml colistin (co-culture populations were treated with colistin in the presence of co-evolving fungal cells). For each population, the average survival rate of 4 replicates is shown (n.s. indicates not significant, Mann–Whitney U test). **(C)** *P. aeruginosa* cells from co-cultured-evolved populations (orange circle) lost colistin resistance upon removal of *C. albicans* with antifungal nystatin treatment (gray circle) or upon $Mg^{2+}$ supplementation (yellow triangle). Mean ± std of 4 biological replicates is shown (* $p < 0.05$, ** $p < 0.01$, Mann–Whitney U test). **(D)** Genome sequencing revealed that *P. aeruginosa* strains that evolved enhanced colistin survival either in monoculture or in co- had different numbers of fixed mutations per population (** $p < 0.01$, Mann–Whitney U test) due to recurrent hypermutator mutations in monoculture populations. **(E)** Monoculture- and co-culture-evolved populations acquired enhanced colistin survival via a distinct spectrum of putative adaptive mutations. Columns indicate each individual replicate monoculture (blue) or co-culture (orange)-evolved population. Rows indicate genes that acquire mutations, grouped by cellular processes. We list all genes with fixed mutations (gray) in 5 or more replicate populations in either condition (* indicates a mutation in the promoter region). The only genes common between monoculture and co-culture-adapted populations are *colS*, previously known for colistin resistance, and *PA3969* (indicated with a #), which is not unique to colistin-treated populations. A more comprehensive list of all fixed mutations is presented in S3 and S4 Tables. The data underlying this Fig 4B–4D can be found in S19 Data. BHI, brain heart infusion; WT, wild-type.

## Discussion

By investigating bacterial defense strategies against fungi, we discovered that $Mg^{2+}$ is a novel, general axis of nutritional competition between fungi and gram-negative bacteria (Fig 5). Under low $Mg^{2+}$ conditions, diverse gram-negative bacteria such as *P. aeruginosa*, *E. coli*, and *S.* Typhimurium rely on a specific $Mg^{2+}$ transporter, MgtA, to overcome fungal-mediated $Mg^{2+}$ sequestration. Despite reducing bacterial fitness, such $Mg^{2+}$ sequestration also shields bacteria from polymyxin antibiotics used for treating multi-drug resistant gram-negative bacteria while impeding the genetic evolution of colistin resistance (Fig 5). Overall, our study uncovers pivotal but previously unexplored connections between nutritional competition, antibiotic survival, and microbial evolution.

Our study indicates that fungal–bacterial competition for $Mg^{2+}$ likely occurs in conditions with $Mg^{2+}$ levels lower than approximately 0.45 mM, a threshold that aligns with lower $Mg^{2+}$ conditions observed during *P. aeruginosa* infection in humans [64]. In addition to fungal sequestration, physiological availability of $Mg^{2+}$ can be affected by various cation-chelating agents during infection, including citrates produced by other bacteria [65], or extra-cellular

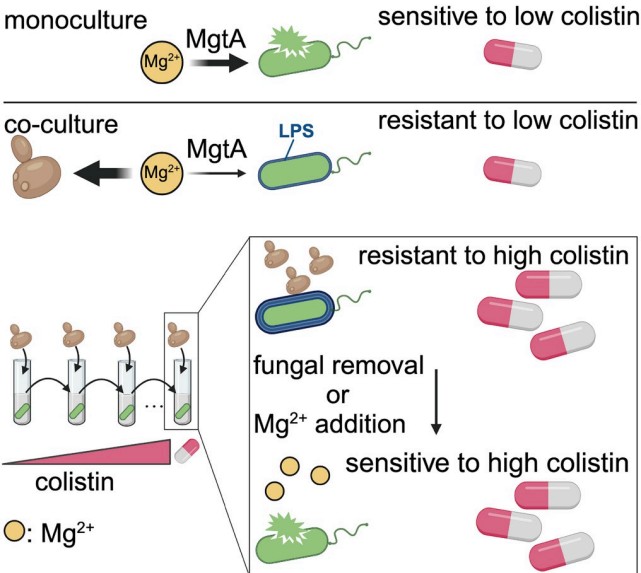

**Fig 5. Mg²⁺ sequestration by fungi suppresses bacterial fitness but profoundly impacts the acquisition of colistin resistance.** We present a model to summarize our findings. *C. albicans* sequesters essential $Mg^{2+}$ ions from *P. aeruginosa*. In turn, the bacterium competes for $Mg^{2+}$ ions using the $Mg^{2+}$ transporter, MgtA. Fungal-mediated $Mg^{2+}$ sequestration alters LPS modification on the bacterial outer membranes and enhances bacterial resistance to colistin. In co-culture with *C. albicans*, *P. aeruginosa* evolves non-canonical fungal-dependent colistin resistance, which can be disrupted by fungal removal or $Mg^{2+}$ supplementation. This figure is created with Biorender.com. LPS, lipopolysaccharide.

DNA in biofilms [66], or macrophage-induced $Mg^{2+}$ sequestration [67]. Low serum $Mg^{2+}$ levels are also associated with other polymicrobial infectious diseases, like diabetic foot ulcers [68] and urinary tract infections [69]. However, understanding microbial access to $Mg^{2+}$ in polymicrobial infections under physiological conditions remains challenging. Although half of cystic fibrosis patients suffer from hypomagnesemia (serum $Mg^{2+}$ levels < 0.75 mM) [70,71], the physiological $Mg^{2+}$ levels in their lungs or sputum are understudied. The few reports of $Mg^{2+}$ levels in CF sputum from clinical studies range from 0.49 ± 0.32 mM [64] to 0.78 to 1 mM [72,73]. This range could reflect differences in patient conditions, sampling methods, or $Mg^{2+}$ quantification techniques. The complexity of CF environments further complicates assessing $Mg^{2+}$ availability for microbes during infections. Moreover, measurements of $Mg^{2+}$ levels based on sputum samples may not reflect the bioavailability of magnesium in different host microenvironments. Despite this uncertainty, molecular evidence suggests that *P. aeruginosa* may indeed experience $Mg^{2+}$ depletion in CF airways. Several *P. aeruginosa* genes that are responsive to low $Mg^{2+}$ [74], including *mgtA* [75], are induced in CF sputum. Other gram-negative bacteria, including *S.* Typhimurium, also encounter reduced $Mg^{2+}$ concentrations upon infection of hosts [67]. We propose that using microbial $Mg^{2+}$ genetic biosensors in animal infection models would be a logical next step to determine bioavailable $Mg^{2+}$ levels to gram-negative bacteria during infection.

Although low $Mg^{2+}$ was previously shown to affect colistin resistance in gram-negative bacteria [52,53], our study provides the first molecular evidence that fungal–bacterial competition for $Mg^{2+}$ may profoundly impact both bacterial fitness and colistin survival. Our study reveals context-dependent fitness effects of $Mg^{2+}$ competition, which incurs general fitness costs to bacteria but provides fitness benefits in terms of colistin resistance. This tradeoff might favor

polymicrobial infections, either in treatment conditions involving polymyxins or when host immune cells produce $Mg^{2+}$-dependent antimicrobial peptides that act upon bacteria. Future exploration into how $Mg^{2+}$ competition alters other antibiotic or antifungal resistance will broaden our understanding of how polymicrobial infections influence the onset of drug resistance. $Mg^{2+}$ has been implicated in altering the susceptibility of gram-negative bacteria to antibiotics other than colistin. For instance, in *S.* Typhimurium, impaired *mgtA* or $Mg^{2+}$ homeostasis increases susceptibility to cyclohexane or nitrooxidative stress [76,77]. Our study highlights the importance of studying how $Mg^{2+}$ homeostasis broadly impacts antimicrobial resistance in gram-negative bacteria.

Our evolution experiments provide novel insights into how co-habitation with *C. albicans* can influence the evolution of *P. aeruginosa*. The fungal- and $Mg^{2+}$-dependent colistin survival we observed has crucial biomedical implications. Disrupting this fungal protection, either via concomitant antifungal therapy or via $Mg^{2+}$ supplementation may enhance the efficacy of colistin treatment or even rescue it even after colistin resistance has been apparently acquired. Consistent with this idea, hypomagnesemia in various chronic infections exacerbates disease progression, whereas oral $Mg^{2+}$ supplementation alleviates it [78,79].

The rise of colistin-resistant bacteria in environments is an escalating concern [80,81]. Our discovery of genetic determinants for fungal–bacterial competition paves the way to investigating whether fungi and bacteria compete for $Mg^{2+}$ during polymicrobial infections, what the consequences of such competition are to microbes and hosts, and how fungal–bacterial interactions alter microbial drug resistance in vivo. Further investigations using *P. aeruginosa* clinical isolates with a history of *C. albicans* co-infection will extend our in vitro findings to clinical contexts and illuminate the role of $Mg^{2+}$ or other factors in shaping fungal–bacterial interactions and drug resistance during co-infection. In addition to polymicrobial infections in animals, low $Mg^{2+}$ is associated with exacerbation of plant infectious diseases [82]. Since our study finds that $Mg^{2+}$ competition is widespread between fungi and gram-negative bacteria, it also hints at the exciting possibility of manipulating environmental $Mg^{2+}$ levels or fungal–bacteria interactions to constrain infectious diseases in diverse eukaryotic hosts, as well as mitigating the spread of colistin resistance. Overall, our findings reveal a novel dimension in nutritional competition and highlight the transformative potential of fundamental discoveries in guiding effective strategies to treat intractable polymicrobial infections or mitigate antibiotic resistance.

## Materials and methods

### Bacterial and fungal strains and culture conditions

All bacterial and fungal strains used in this study are listed in S5 Table in the supplemental material. Bacterial strains in this study were derived from *P. aeruginosa* PAO1, *E. coli* BW25113, *S.* Typhimurium 14028S, *S. aureus* SA113, and *B. multivorans* AMT17616. Yeast strains were derived from *C. albicans* SC5314, *S. cerevisiae* W303. *C. tropicalis* CBS94, *C. parapsilosis* CBS604, and *C. glabrata* CBS562 were purchased from ATCC. Individual *P. aeruginosa* Tn mutant strains were obtained from Dr. Colin Manoil's lab [28,54]; correct transposon insertions were confirmed by PCR and Sanger sequencing. Clinical *P. aeruginosa* isolates from CF patients were obtained from Dr. Pradeep Singh's lab. Unless otherwise specified, all experiments were performed in Brain Heart Infusion Broth Media (BHI, Sigma-Aldrich), buffered with 10% MOPS to pH 7.0 followed by filter sterilization. All strains were grown in BHI and incubated at 30 ˚C or 37 ˚C. Other media used in this study were YPD (1% yeast extract, 2% peptone, and 2% D-glucose), SCFM [39], and TSB (Sigma-Aldrich). For antibiotics used in this study, 100 μg/ml gentamicin was used to select against gram-negative bacteria, and 50 μg/ml nystatin was used to select against *Candida* species and *S. cerevisiae*. Colistin and

polymyxin B (Sigma-Aldrich) were prepared as 10 mg/ml and 5 mg/ml stock solutions, respectively.

## Generation of *P. aeruginosa* mutants

For generating gene deletions in *P. aeruginosa*, a pEXG2-mediated allelic exchange method was used [83]. In short, pEXG2 deletion constructs were transformed into *E. coli* S17. S17 donor was subsequently mixed with *P. aeruginosa* recipients on an LB agar plate at a 5:1 ratio of donor to recipient cells, and the cell mixture was incubated at 30 ˚C overnight. The cell mixture was then scraped, resuspended into 200 μl LB, and plated on LB agar plates with 100 μg/ml gentamicin to select for cells containing the deletion plasmid integrated into the *P. aeruginosa* genome. *P. aeruginosa* merodiploid colonies were streaked on LB-no salt-agar plates with sucrose. Gentamicin-sensitive and sucrose-resistant colonies were screened for allelic replacement by colony PCR, and gene deletions were confirmed by Sanger sequencing of PCR products. Primers and plasmids used for strain constructions are listed in S6 Table.

## Generation of *S. cerevisiae* mutants

*S. cerevisiae* gene deletion mutants were generated by homologous recombination at endogenous gene loci. Gene deletion fragments were amplified from pFA6a-natMX4 and fused with 500 bp upstream and downstream sequences of the targeted coding sequence by PCR. The PCR fragment with homology arms was transformed into *S. cerevisiae* cells with the standard LiOAc-based protocol [84]. Gene deletions were confirmed by colony PCR and Sanger sequencing to check the absence of endogenous genes. Primers and plasmids used for strain constructions are listed in S6 Table.

## Fungal–bacterial co-culture CFU assay

All experiments began with bacterial or fungal cultures that were grown overnight. The starter cultures were diluted either 1:100 (for bacterial cultures) or 1:50 (for fungal cultures) in BHI and cultured for approximately 4 to 5 h to reach the log phase. Refreshed bacterial and fungal strains were added to BHI to reach a final bacterial cell density of $2.5 \times 10^4$ cells/ml and a fungal cell density of $5 \times 10^5$ cells/ml, for the co-culture experiment. The same amounts of bacterial or fungal cells were also added separately in fresh BHI, as monoculture controls. These cultures were incubated for 18 h at 30 ˚C or 37 ˚C with shaking. We chose 30 ˚C for the initial co-culture assays for 2 reasons. First, *C. albicans* cells reached higher CFU at 30 ˚C than 37 ˚C, which would impose a stronger competition with bacteria. Second, *C. albicans* cells form hyphae at 37 ˚C, which can have multiple cells in one filament and thus confound CFU measurements. We further confirmed that our findings of $Mg^{2+}$ competition are independent of temperature by setting up co-culture assays at both 30 ˚C and 37 ˚C. Cultures were serially diluted and spotted on LB plates with 50 μg/ml nystatin for measuring bacterial growth and on YPD plates with 100 μg/ml gentamicin for measuring fungal growth.

## *PA4824* or *mgtA* complementation assay

To complement the deletion of *mgtA*, Δ*mgtA* was transformed with a genome-integrated copy of *mgtA* under its endogenous promoter (*miniTn7-mgtA*) [85] or a plasmid copy of *mgtA* under an arabinose-inducible promoter (p*BAD*) [86]. As *PA4824* is in the same operon as *mgtA*, we used p*BAD* to express *PA4824* to ensure its full expression. These deletion-complemented strains were prepared in the same way as the co-culture CFU assay, except that 0.2% arabinose was used to induce *PA4824* or *mgtA* expression.

## Bacterial–bacterial co-culture CFU assay

Overnight bacterial cultures were diluted 1:100 in BHI and cultured for approximately 4 to 5 h to reach log phase. Log-phase cultures of bacterial competitors (*S. aureus*, *S. multivorans*, *E. coli*, or *S.* Typhimurium) were added in fresh BHI at $5 \times 10^5$ cells/ml as final cell density while $2.5 \times 10^4$ cells/ml of log-phase *P. aeruginosa* culture was added. The same number of bacterial cells were also added separately in fresh BHI, as monoculture controls. These cultures were incubated for 18 h at 30 ˚C with shaking. Cultures were serially diluted and spotted on Pseudomonas Isolation Agar (Millipore-17208) to measure *P. aeruginosa* CFUs.

## Measurements of bacterial intracellular Mg$^{2+}$

Cytosolic Mg$^{2+}$ concentrations in *P. aeruginosa* were measured using a modification of the previously reported Mg$^{2+}$ genetic sensor assay [31]. First, we cloned the Mg$^{2+}$ genetic sensor containing *S.* Typhimurium *mgtA* leader sequence ($P_{lac}$-*mgtA$_{leader}$*-*lacZ*) or the promoter-only fragment ($P_{lac}$-*lacZ*) into a *P. aeruginosa* replicating plasmid, pBBR [87]. Then, these 2 plasmids were transformed into the indicated *P. aeruginosa* strains separately, with the former acting as a reporter for Mg$^{2+}$ and the latter acting as a control. Both strains were cultured in BHI or fungal spent media (80% v/v final concentration) containing 100 μg/ml gentamicin for 16 h and then diluted 1:100 in the corresponding media with gentamicin for 4 h to reach log phase. The cell density of these cultures was adjusted to OD600 = 0.4, and we followed the same protocol to measure β-galactosidase activity using Galacto-Light Plus System (Thermo Fisher). Finally, cytosolic Mg$^{2+}$ concentrations were derived from the following equation, where the control strain was used to normalize transcription or translation efficiency between strains:

$$[\mathrm{Mg}^{2+}] = (\text{miller units from the control strain})/(\text{miller units from the reporter strain})$$

## Colistin survival assay in *C. albicans*-spent media

To prepare *C. albicans*-free supernatant, a single *C. albicans* colony was inoculated in 10 ml BHI and cultured at 30 ˚C for 24 h. The fungal culture was centrifuged, and the supernatant was filtered with a Steriflip unit with a 0.22 μm Millipore Express PLUS membrane (Millipore-Sigma), and 8 ml fungal filtrate was added to 2 ml fresh BHI (80% v/v final concentration) to prepare *C. albicans*-spent media, mimicking nutrient exhaustion by *C. albicans*. To perform colistin treatment, we first diluted overnight bacterial culture in 3 ml BHI or *C. albicans*-spent media at 1:100 for 4 h. Then, bacterial cultures were adjusted to OD600 = 0.3 in BHI or *C. albicans*-spent media. Each bacterial culture was treated with colistin at the indicated concentration, and the samples were incubated at 30 ˚C for 1.5 h. To assay bacterial viability, bacterial cultures after colistin treatment were serially diluted on LB plates. Bacterial cultures in the same growth condition without colistin treatment were used as a control. Bacterial survival was calculated by the ratio of bacterial CFUs, relative to those observed without treatment, at 1.5 h.

## Colistin survival assay in co-culture

Log-phase bacterial cells were cultured in BHI alone or in co-culture with fungal cells in BHI for 18 h at 30 ˚C or 37 ˚C, as the co-culture CFU assay indicated. To equilibrate bacterial cell numbers in these 2 culturing conditions prior to the colistin treatment, the cell density of monoculture samples was adjusted to OD600 = 0.3 to match the bacterial cell density after 18-h growth in co-culture. Both equilibrated monoculture and co-culture samples were split

into 1 ml in 2 different tubes for colistin-treated versus untreated conditions. Colistin was added to the first tube at the indicated concentration. Bacterial cells in both tubes were incubated at 30 ˚C for 1.5 h and the bacterial viability was determined by counting CFUs after serial dilution. Bacterial survival was determined by the ratio of bacterial CFUs upon colistin treatment, relative to without treatment (average of 4 replicates). Bacterial survival was set to 100% if it was over 100% due to stochastic variation in colony counts.

We used the same fungal protection assay to measure colistin resistance in co-culture-evolved populations. These evolved populations were split into 2 culturing conditions: BHI only, as co-culture conditions, or BHI + 50 μg/ml nystatin, to remove fungal cells (as monoculture conditions). Cells were grown for 18 h at 37 ˚C. After overnight growth, populations in both conditions were treated with 192 μg/ml colistin following the above protocol. To measure bacterial viability after colistin treatments, cells in co-culture and monoculture conditions were spotted on LB + 50 μg/ml nystatin or LB plates, respectively.

### Competitive fitness assay

A WT PAO1 strain with chromosomally integrated miniTn7-mCherry was used as the reference strain in fitness competition with deletion mutants, either in BHI only or in co-culture with *C. albicans*. All bacterial and yeast strains were incubated in BHI at 30 ˚C to log phase, and cell density was quantified by OD600 using a spectrophotometer (biowave CO8000 Cell Density Meter). First, a non-fluorescent strain, either the PAO1 wild-type or deletion mutants, was mixed with the fluorescence-labeled reference strain at a ratio of 1:1 to make a bacterial mixed culture with $2 \times 10^6$ cells/ml. Part of the cell mixture was used to determine the initial ratio of sample and reference strains using flow cytometry (BD FACSymphony A5 Cell Analyzer). Then, 10 μl of cell mixture was inoculated separately in 3 ml BHI alone (monoculture) or 3 ml BHI with $2 \times 10^5$ *C. albicans* cells (co-culture) and grown in BHI at 30 ˚C for 18 h. After 18 h, monoculture samples were diluted 100-fold to measure the ratio of sample and reference strain. For co-culture samples, a low-speed spin (1,000x*g* for 3 min) was applied to separate bacterial and fungal populations. Supernatants enriched with bacterial cells were measured using flow cytometry. For each sample, at least 30,000 cells were collected and the FACS data were analyzed by the FlowJo 10.4.1 software. The reference strain was cultured separately to estimate the number of generations during the experiment. Each experiment was conducted in at least 2 biological and 2 technical replicates. To calculate the relative fitness, *w*, of each sample strain to the reference strain, we followed the formula: $w = 1+s$, where selection coefficient s = [ln(sample/reference)$_t$- in(sample/reference)$_{t = 0}$]/t, where t=number of generations and (sample/reference) is the ratio between a sample strain and the reference strain [88].

### RNA sequencing and analysis

A log-phase *P. aeruginosa* WT culture (OD600 = 0.1~0.2) was diluted at $5 \times 10^6$ cells in 100 ml media alone or 100 ml media with $5 \times 10^7$ *C. albicans* cells and grown at 30 ˚C. Cells were collected for extracting RNA at 8 h when there was no significant growth difference of *P. aeruginosa* between monoculture and co-culture. Prior to RNA extraction, a slow-speed spin was applied to co-culture samples to enrich bacteria in the supernatant. For each sample, total RNA was harvested from $10^9$ cells, preserved in RNA-protected QIAzol, and purified by RNAeasy mini kit (Qiagen) following the commercial protocol. RNA was quantified by Qubit fluorometric quantitation system (Thermo Fisher), and RNA integrity was examined by 4200 TapeStation System (Agilent). We generated an rRNA depleted library and obtained >20 million 150 bp pair-end Illumina NovaSeq reads for each sample commercially (Azenta). Reads were examined using FastQC (Barbaham Bioinformatics, Babraham Institute) and trimmed

by Trim Galore! (v0.6.10; https://github.com/FelixKrueger/TrimGalore). Reads were mapped to the *P. aeruginosa* PAO1 genome (GCF_00006765.1) using Subread/FeatureCounts (https://github.com/ShiLab-Bioinformatics/subread). Differential gene expression was analyzed with DESeq2 [89] (https://bioconductor.org/packages/release/bioc/htmL/DESeq2.htmL), following the default pipeline of data normalization. A $\log_2$ fold change cutoff of 2 and a false discovery rate (FDR) of 0.05 were used to filter significantly differentially expressed genes between samples from 3 biological replicates. Gene Ontology analysis was done using the *P. aeruginosa* STRING database on ShinyGO 0.77(http://bioinformatics.sdstate.edu/go/).

Sequencing files are stored in the BioProject PRJNA1021673 of the NCBI repository.

### Transposon-sequencing and analysis

We used a pool of ~$10^5$ transposon-insertion mutants, generated by Tn5-based transposon T8 (IS*lacZ*hah-tc) insertion in the *P. aeruginosa* strain PAO1 [28], and 20 μl of frozen Tn mutants (approximately containing $10^8$ cells) was thawed and inoculated in 50 ml media or 50 ml media with $10^8$ *C. albicans* cells in a 250-ml flask, to establish monoculture and co-culture conditions, each with 3 biological replicates. To screen the fitness of Tn mutants effectively, cells were grown at 30 ˚C with shaking at 210 rpm for 10 h (approximately 10 generations) and collected for genomic DNA extraction using the DNAeasy Blood & Tissue kit (Qiagen). DNA samples were prepared for transposon sequencing following the previously published TdT/two-step PCR method [90]. In short, 2 μg DNA was end-repaired by NEB end-repair reaction and subsequently C-tailed in the TdT tailing reaction. Two PCR reactions were used to enrich DNA fragments spanning the junction of transposon insertion. PCR-1 was performed using a transposon-specific primer and a polyG primer (olj376) on 0.5 to 1 μg DNA; at least 2 reactions which were pooled in the following purification steps. For PCR-2, a mixture of Tn8 primers combined with 3N, 5N, 7N, and 9N random bases were used to increase the diversity of Tn-seq libraries. The size of library DNA was examined by 4200 High Sensitivity DNA TapeStation System (Agilent) and quantified using qPCR with standard Illumina primers (Kapa Biosystems). Libraries were pooled and sequenced commercially using an Illumina NovaSeq or NextSeq sequencer (Novogene). For each sample, at least 25 million pair-end 150 bp sequencing reads were generated after which, we proceeded for downstream analysis. Sequencing files are stored in the BioProject PRJNA1021673 of the NCBI repository.

The analysis scripts used in this study are adapted from previously published protocols and are available in the GitHub repository (https://github.com/PhoebeHsieh-yuying/P.-aeruginosa_Tnseq_paper) and the Zenodo repository (DOI: 10.5281/zenodo.11404260). Briefly, sequencing reads were groomed by FastQC (Barbaham Bioinformatics, Babraham Institute), filtered to keep those with the correct Tn8 sequence (TATAAGAGTCAG) near the 5′ end using Cutadapt [91] and mapped to the *P. aeruginosa* PAO1 genome (GCF_00006765.1) with Bowtie2 [92] using the Tnseq2.sh script. The output table with transposon insertion site assignments of each sample was concatenated and annotated with known gene features of *P. aeruginosa* using a custom R script, Tnseq_reformat.R. Then, DESeq2 [89] statistical package in R was used to identify genes with significant changes in Tn insertion between co-culture and monoculture. A $\log_2$ fold change cutoff of 1 and an FDR of 0.05 were used as the criteria for significance.

### Experimental evolution of colistin survival

Eight independent colonies of a WT *P. aeruginosa* strain PAO1 were used as ancestral strains for the experimental evolution experiment. Each ancestral clone was inoculated and grown in BHI at 37 ˚C to $10^9$ cells/ml. At the beginning of the experiment, approximately $10^7$ cells of

each ancestral population were transferred either into 1 ml BHI with colistin or into 1 ml BHI with colistin and $10^6$ *C. albicans* cells to select colistin resistance. Each day, populations were passaged at a ratio of 1:200 in the former condition, while co-cultured populations were passaged at 1:50 in colistin-containing BHI to maintain comparable population size between both conditions. The concentration of colistin was started from 1.5 μg/ml, gradually increased 2-fold every week or every 2 weeks until it reached 192 μg/ml at the end of the 90-day evolution experiment. As controls for adaptation to BHI only or co-culture with *C. albicans*, the same ancestral populations were passaged at a ratio of 1:200 into 1 ml BHI or 1 ml BHI with the same amount of *C. albicans* cells, but without colistin. After every 7 days, 1 ml evolving cultures were mixed with 500 μl 80% glycerol and frozen at −80 ˚C. On day 90, we expected that populations evolved in colistin-containing BHI with *C. albicans* had been passaged for approximately 600 generations, while the rest of the evolved populations had been passaged for approximately 700 generations.

### Whole-genome sequencing and analysis

We sequenced 2 ancestral populations and all 8 evolved populations per treatment. We revived each population from a freezer stock in the growth condition under which they evolved and grew for 24 h. For co-culture evolved populations, we treated them with nystatin to remove fungal cells, and 3 ml bacterial culture was collected for DNA extraction using the DNAeasy Blood & Tissue kit (Qiagen). Sequencing libraries were made and sequenced commercially by Illumina sequencers in SeqCenter (https://www.seqcenter.com/). Variant calling was done using the breseq [93] software v0.37.1 and the *P. aeruginosa* PAO1 genome (GCF_00006765.1) was used as the reference genome. The average depth of sequencing for populations was 222.5 fold (+/− 32.7) and the average genome coverage was 98.8 fold (+/− 0.04). Variants at >95% in each population were filtered for mutation calling. Mutations were manually curated by identifying unique variants in each evolved population compared to the ancestor.

Sequencing files are stored in the BioProject PRJNA1021673 of the NCBI repository.

### $Mg^{2+}$ measurements in media

$Mg^{2+}$ concentrations were measured using the Magnesium Assay kit (Sigma-Aldrich) according to the protocol provided. Absorbances at $OD_{450}$ after the enzymatic reaction were compared to a standard curve of $Mg^{2+}$ concentration to determine the absolute concentration in media.

## Supporting information

**S1 Fig. *P. aeruginosa* fitness is suppressed by *C. albicans* in BHI.** During co-culture in BHI for 42 h, *P. aeruginosa* fitness was significantly impaired relative to monoculture, by nearly 100-fold at 20 h. In contrast, *C. albicans* fitness was similar in monoculture versus co-culture (see Methods). The fitness of *C. albicans* or *P. aeruginosa* is shown as colony-forming units (CFUs) per ml in (A) or (B), respectively. Mean ± std of 3 biological replicates is shown (* $p < 0.05$, unpaired two-tailed Student's *t* test used). The data underlying this figure can be found in S1 Data.
(EPS)

**S2 Fig. *P. aeruginosa* fitness remains unchanged during the first 8 h in co-culture with *C. albicans*.** The WT *P. aeruginosa* strain was grown in BHI media only or in co-culture with *C. albicans* (see Methods). The fitness of *P. aeruginosa* was examined every 2 h in BHI (blue) and co-culture (orange) by measuring CFU per ml. *P. aeruginosa* fitness was not significantly

different in monoculture versus co-culture for the first 8 h but began to significantly diverge by the 10-h time point. Mean ± std of 3 biological replicates is shown (** $p < 0.01$ and n.s. indicates not significant. Unpaired two-tailed Student's $t$ test was used, relative to monoculture at each time point). The data underlying this figure can be found in S2 Data.
(EPS)

**S3 Fig. Differentially expressed *P. aeruginosa* genes in co-culture relative to monoculture conditions.** Gene Ontology analysis of genes up-regulated (A) or down-regulated (B) in co-culture compared to monoculture is shown. Up-regulated genes were enriched for iron uptake and RNA polymerase sigma factor 70, whereas down-regulated genes were enriched for bacterial secretion, co-factor biosynthesis, and energy generation. X-axis indicates fold enrichment of genes of interest, and Y-axis indicates GO categories, which are ordered and color-coded by the inverse of their false discovery rate. The number of genes of interest in each category is labeled with the size of circles. (C) RNA-seq volcano plot (see Methods) shows *P. aeruginosa* genes that were differentially expressed in co-culture versus monoculture conditions. X-axis indicates fold change, while Y-axis indicates $p$-value after correcting for multiple testing. $|\log_2$ fold change$| > 2$ and adjusted $p < 0.05$ was used as statistical cutoffs. Up-regulated genes are labeled in red, while down-regulated genes are in blue. The data underlying this figure can be found in S3 Data.
(EPS)

**S4 Fig. Competitive fitness assay verifies the fitness effect of *PA4824* or *mgtA* mutants in co-culture.** (A) Schematic of the fluorescence-based competitive fitness assay that mimics bulk selection in Tn-seq (see Methods). (B) Relative fitness of *PA4824* or *mgtA* mutants to the WT *P. aeruginosa* strain was impaired in co-culture conditions. Relative fitness in BHI media only is indicated in blue, and relative fitness in co-culture with *C. albicans* is indicated in orange. Mean ± std of 3 biological replicates is shown (** $p < 0.01$, unpaired two-tailed Student's $t$ test used). (C) A WT copy of *mgtA* restored the fitness of Δ*mgtA* mutant in co-culture. Mean ± std of 4 biological replicates is shown (** $p < 0.01$, n.s. indicates not significant, Dunnett's one-way ANOVA test was used). (D) The fitness of Δ*PA4824* or Δ*mgtA* mutant was significantly restored by arabinose-induced expression of *PA4824* or *mgtA*. Δ*PA4824* was transformed with pJN105-*PA4824* and Δ*mgtA* was transformed with pJN105-*mgtA*. Deletion mutants with pJN105 (vector only) were used as controls. Mean ± std of 4 biological replicates is shown (** $p < 0.01$, unpaired two-tailed Student's $t$ test was used). The data underlying S4B–S4D Fig can be found in S5 Data.
(EPS)

**S5 Fig. Verification of the Mg²⁺ genetic sensor in *P. aeruginosa*.** The WT *P. aeruginosa* strain carrying the $Mg^{2+}$ genetic sensor (pBBR-$P_{lac1-6}$-$mgtA_{leader}$-$lacZ$) was cultured in minimal medium with 10 μm or 10 mM $Mg^{2+}$ to log phase, mimicking low or high $Mg^{2+}$ conditions. Relative concentrations of intracellular $Mg^{2+}$ were determined by the measurements of β-galactosidase activity (see Methods). Mean ± std of 3 biological replicates is shown (** $p < 0.01$, unpaired two-tailed Student's $t$ test used). The data underlying this figure can be found in S7 Data.
(EPS)

**S6 Fig. Alphafold structural prediction suggests PA4824 may function as a Mg²⁺ transporter.** (A) An Alphafold-predicted structure of PA4824. Confidence in structural prediction is shown by color, where dark and light blue indicate high confidence. (B) and (C) show the top-down view of the core of PA4824, displayed by Chimerax. In (B), residues are labeled by their hydrophobicity (green: hydrophilic residues, yellow: hydrophobic residues). In (C), residues are labeled by their electrostatic features (red: negatively charged residues, blue: positively

charged residues).
(EPS)

**S7 Fig. Two other *P. aeruginosa* Mg$^{2+}$ transporters, CorA and MgtE, are not involved in Mg$^{2+}$ competition with *C. albicans*.** (A) Expression levels of *mgtE* and *corA* were unchanged in response to *C. albicans* presence. RNA-seq results of *mgtE*, *corA*, and *mgtA* across 4 experimental conditions (BHI only, co-culture in BHI, BHI with supplemental Mg$^{2+}$ (10 mM), and co-culture in BHI supplemental Mg$^{2+}$ (10 mM)) are shown in z-score. The log$_2$ fold change of each gene in co-culture versus monoculture in BHI is shown on the right. (B) *corA* or *mgtE* Tn mutants did not confer fitness cost in co-culture with *C. albicans*, whereas Δ*mgtA* mutant did. Fitness of *P. aeruginosa* strains in BHI only or co-culture conditions are shown in blue or orange, respectively. Mean ± std of 4 biological replicates is shown (** $p < 0.01$, * $p < 0.05$, and n.s. indicates not significant. Dunnett's one-way ANOVA test was used). (C) *corA or mgtE* mutants did not show reduced intracellular Mg$^{2+}$ levels in *C. albicans* spent BHI, whereas Δ*mgtA* mutants did. The same experimental setup in Fig 1E was used for this assay. Mean ± std of 3 biological replicates is shown (** $p < 0.01$, Dunnett's one-way ANOVA test was used). The data underlying S7A–S7C Fig can be found in S8 Data.
(EPS)

**S8 Fig. Mg$^{2+}$ supplementation rescues the fitness effects of several candidate genes, without altering their expression.** (A) Tn-seq volcano plot in Mg$^{2+}$-repleted conditions (see Methods) shows the fitness effects of most of the candidate defense and dispensable genes (in black) were restored (compare to Fig 1B). X-axis indicates fold change, while Y-axis indicates *p*-value after correcting for multiple testing. We used log$_2$ fold change $> 2$ and adjusted *p*-value $< 0.1$ as statistical cutoffs. *PA4824*, *PA4825*, and *PA4826* are labeled in yellow. Under this condition, only 1 gene was found to be dispensable (in red, *PA2854*), and 2 genes were found to be essential (in blue, *PA5484* and *PA4029*). (B) Mg$^{2+}$ supplementation only altered the expression of *PA4824*, *PA4825*, and *PA4826* in co-culture, but not the other candidate genes. Heatmap shows the expression of 18 dispensable genes and 8 fungal-defense genes in co-culture versus monoculture, with or without Mg$^{2+}$ supplementation. The data underlying S8A and S8B Fig can be found in S9 Data.
(EPS)

**S9 Fig. Mg$^{2+}$-sequestration by fungi depends on environmental Mg$^{2+}$ levels and primarily determines bacterial fitness in co-culture.** (A–C) Fitness of *P. aeruginosa* Δ*mgtA* mutant was measured in co-culture with *C. albicans* in TSB (A), SCFM (B), or YPD (C). The Δ*mgtA* mutant had impaired fitness in TSB (like in BHI media), but not SCFM or YPD media. Bacterial fitness in media only or in co-culture is shown in blue or orange, respectively. Mean ± std of 4 biological replicates is shown (** $p < 0.01$ and n.s. indicates not significant. Unpaired two-tailed Student's *t* test was used). (D) BHI and TSB exhibited lower Mg$^{2+}$ levels compared to YPD and SCFM, both in the fresh media and in *C. albicans* spent media. Mean ± std of 3 biological replicates is shown. (E) Intracellular Mg$^{2+}$ levels in *P. aeruginosa* were lower in the fungal spent media from BHI and TSB but not in fungal spent media from SCFM or YPD. Mean ± std of 4 biological replicates is shown. (F) Fitness of the *P. aeruginosa* Δ*mgtA* mutant in co-culture was rescued by adding more than 0.3 mM Mg$^{2+}$, leading to a total of 0.45 mM Mg$^{2+}$ in BHI. (G) Supplementation of other cations (Zn$^{2+}$, Cu$^{2+}$, Mo$^{6+}$, or Mn$^{2+}$) in co-culture did not restore the fitness of *P. aeruginosa* Δ*mgtA* mutant. Mean ± std of 4 biological replicates is shown (** $p < 0.01$; unpaired two-tailed Student's *t* test was used). (H) Fitness of *P. aeruginosa* Δ*mgtA* mutant in co-culture could only be rescued by supplementation of 10 mM Ca$^{2+}$. Mean ± std of 4 biological replicates is shown (n.s. indicates not significant. Unpaired two-tailed Student's *t*

test was used). The data underlying S9A and S9B Fig can be found in S10 Data.
(EPS)

**S10 Fig. The fitness of *S. cerevisiae mnr2Δ* mutant is similar to WT *S. cerevisiae* in mono-culture and co-culture with *P. aeruginosa*.** *S. cerevisiae* strains (WT is shown in black and *mnr2Δ* in an empty circle) were grown in BHI only or in co-culture with *P. aeruginosa* strains, as described in Fig 2G. Mean ± std of 4 biological replicates is shown (n.s. indicates not significant. Unpaired two-tailed Student's *t* test was used). The data underlying this figure can be found in S12 Data.
(EPS)

**S11 Fig. Bacterial CFU counts of colistin survival assays in *C. albicans* spent BHI (A) *P. aeruginosa* CFU counts in Fig 3C (B) *S.* Typhimurium CFU counts in Fig 3D.** Bacterial CFU before colistin treatment is shown in gray, and after colistin treatment in orange. The data underlying S11A and S11B Fig can be found in S14 Data.
(EPS)

**S12 Fig. *C. albicans*-mediated $Mg^{2+}$ sequestration confers resistance to polymyxin B by *P. aeruginosa*.** The WT *P. aeruginosa* strain was highly susceptible to 2.5 μg/ml polymyxin B in BHI, became resistant in *C. albicans*-spent BHI, but became susceptible again in *C. albicans*-spent BHI supplemented with 10 mM $Mg^{2+}$. Mean ± std of bacterial survival from 4 biological replicates is shown (** $p < 0.01$, Dunnett's one-way ANOVA test used). The data underlying this figure can be found in S15 Data.
(EPS)

**S13 Fig. Bacterial CFU counts of colistin survival assays in co-culture with *C. albicans*.** (A) *P. aeruginosa* CFU counts in Fig 3F. (B) *S.* Typhimurium CFU counts in Fig 3G. Bacterial CFU in monoculture is shown in blue and co-culture in orange. The data underlying S13A and S13B Fig can be found in S16 Data.
(EPS)

**S14 Fig. *C. albicans* protects both *P. aeruginosa* and *S.* Typhimurium from colistin at 37 ˚C.** (A) *P. aeruginosa* in co-culture with *C. albicans* was more resistant to colistin, compared to monoculture, at 37 ˚C (** $p < 0.01$, unpaired two-tailed Student's *t* test used). (B) *S.* Typhimurium in co-culture with *C. albicans* was more resistant to colistin, compared to monoculture, at 37 ˚C (** $p < 0.01$, unpaired two-tailed Student's *t* test used). (C) *P. aeruginosa* CFU counts in S14A Fig. (D) *S.* Typhimurium CFU counts in S14B Fig. Bacterial CFU in monoculture is shown in blue and co-culture in orange. The data underlying S14A–S14D Fig can be found in S17 Data.
(EPS)

**S15 Fig. *C. albicans* protects *P. aeruginosa* CF isolates from colistin at 37 ˚C.** (A) Eight isolates from 2 sputum samples (CF-S0089 and CF-S1045) were treated with 3 μg/ml colistin, either in monoculture or in co-culture with *C. albicans*, as indicated in colistin survival assay (see Methods). (A) Bacterial survival is shown as the percentage of viable cells after the colistin treatment. Mean ± std of 3 biological replicates is shown (** $p < 0.01$, * $p < 0.05$, unpaired one-tailed Student's *t* test used). (B) *P. aeruginosa* CFU counts of experiments in the panel A. Bacterial CFU in monoculture (light blue), monoculture with 3 μg/ml colistin (dark blue), co-culture (brown), and co-culture with 3 μg/ml colistin (orange) are shown here. The data underlying S15A and S15B Fig can be found in S18 Data.
(EPS)

**S16 Fig. Evolution of *P. aeruginosa* in BHI.** WT *P. aeruginosa* populations were passaged daily without colistin, to assess bacterial adaptation to BHI or co-culture with *C. albicans* in the absence of colistin.
(EPS)

**S17 Fig. CFU counts of the co-culture evolved populations of experiments in Fig 4C.** (A) *P. aeruginosa* CFU with (brown) or without (orange) 192 μg/ml colistin before fungal removal. (B) *P. aeruginosa* CFU with (brown) or without (yellow) 192 μg/ml colistin after $Mg^{2+}$ supplementation in co-culture (C) *P. aeruginosa* CFU with (brown) or without (gray) 192 μg/ml colistin after fungal removal.
(EPS)

**S18 Fig. Other putative genetic targets for colistin resistance.** Individual genes that have fixed mutations in more than two of colistin-resistant monoculture or co-culture-evolved populations are listed. Columns refer to individual evolved populations, and rows refer to mutated genes grouped by cellular process. Genes known to be involved in colistin resistance are labeled in red. Gray or blank squares indicate the presence or absence of mutation, respectively (* indicates a mutation in the promoter region).
(EPS)

**S1 Table. *P. aeruginosa* differentially expressed genes in co-culture versus monoculture, in BHI or BHI supplemented with 10 mM $Mg^{2+}$.** The data underlying this figure can be found in S4 Data.
(XLSX)

**S2 Table. *P. aeruginosa* indispensable gene candidates identified in Tn-seq.**
(XLSX)

**S3 Table. Fixed mutations in each colistin-resistant evolved population.**
(XLSX)

**S4 Table. Fixed mutations in each control population without colistin treatment.**
(XLSX)

**S5 Table. Bacterial and fungal strains used in this study.**
(XLSX)

**S6 Table. Primers and plasmids used in this study.**
(XLSX)

**S1 Data. The data underlying S1 Fig.**
(XLSX)

**S2 Data. The data underlying S2 Fig.**
(XLSX)

**S3 Data. The data underlying S3 Fig.**
(XLSX)

**S4 Data. The data underlying S1 Table.**
(XLSX)

**S5 Data. The data underlying S4B–S4D Fig.**
(XLSX)

**S6 Data. The data underlying Fig 1D–1F.**
(XLSX)

**S7 Data. The data underlying S5 Fig.**
(XLSX)

**S8 Data. The data underlying S7A–S7C Fig.**
(XLSX)

**S9 Data. The data underlying S8A and S8B Fig.**
(XLSX)

**S10 Data. The data underlying S9A and S9B Fig.**
(XLSX)

**S11 Data. The data underlying Fig 2A–2G.**
(XLSX)

**S12 Data. The data underlying S10 Fig.**
(XLSX)

**S13 Data. The data underlying Fig 3C, 3D, 3F and 3G.**
(XLSX)

**S14 Data. The data underlying S11A and S11B Fig.**
(XLSX)

**S15 Data. The data underlying S15A and S15B Fig.**
(XLSX)

**S16 Data. The data underlying S13A and S13B Fig.**
(XLSX)

**S17 Data. The data underlying S14A–S14D Fig.**
(XLSX)

**S18 Data. The data underlying S15A and S15B Fig.**
(XLSX)

**S19 Data. The data underlying Fig 4B–4D.**
(XLSX)

## Acknowledgments

We thank Dr. Larry Gallagher (Joseph Mougous lab) and Dr. Gina Lewin (Marvin Whiteley lab) for their generous advice on library construction and data analysis of Tn-seq experiments. We thank Dr. Eduardo Groisman for sharing *S*. Typhimurium strains and $Mg^{2+}$ reporter plasmids. We thank Dr. Colin Manoil and Dr. Pradeep Singh for sharing *P. aeruginosa* strains and Dr. Lauren Ames for sharing *C. albicans* strain. We thank Nicole Smalley for sharing her expertise in *P. aeruginosa* genetics. We thank María Angélica Bravo Núñez, E. Peter Greenberg, Carrie Harwood, Meng-Chao Yao, and members of the Dandekar and Malik labs (Ching-Ho Chang, Peter Dietzen, Rechel Geiger, Grant King, Isabel Mejia Natividad, Samantha Wellington Miranda, Maria Toro Moreno), who provided constructive feedback on our study and manuscript.

Cartoon diagrams in this study were prepared by BioRender (https://www.biorender.com/).

## Author Contributions

**Conceptualization:** Yu-Ying Phoebe Hsieh, Ajai A. Dandekar, Harmit S. Malik.

**Data curation:** Yu-Ying Phoebe Hsieh, Wanting Sun.

**Formal analysis:** Yu-Ying Phoebe Hsieh, Wanting Sun, Janet M. Young, Ajai A. Dandekar, Harmit S. Malik.

**Funding acquisition:** Yu-Ying Phoebe Hsieh, Deborah A. Hogan, Ajai A. Dandekar, Harmit S. Malik.

**Investigation:** Yu-Ying Phoebe Hsieh, Wanting Sun, Janet M. Young, Robin Cheung.

**Methodology:** Yu-Ying Phoebe Hsieh, Wanting Sun, Janet M. Young, Robin Cheung, Deborah A. Hogan, Ajai A. Dandekar, Harmit S. Malik.

**Project administration:** Yu-Ying Phoebe Hsieh, Deborah A. Hogan, Ajai A. Dandekar, Harmit S. Malik.

**Resources:** Ajai A. Dandekar.

**Software:** Janet M. Young.

**Supervision:** Yu-Ying Phoebe Hsieh, Harmit S. Malik.

**Validation:** Yu-Ying Phoebe Hsieh, Wanting Sun, Robin Cheung.

**Visualization:** Yu-Ying Phoebe Hsieh, Wanting Sun, Ajai A. Dandekar, Harmit S. Malik.

**Writing – original draft:** Yu-Ying Phoebe Hsieh, Wanting Sun, Ajai A. Dandekar, Harmit S. Malik.

**Writing – review & editing:** Yu-Ying Phoebe Hsieh, Wanting Sun, Janet M. Young, Deborah A. Hogan, Ajai A. Dandekar, Harmit S. Malik.

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
