## [Editor Report · Decision Letter 0]

5 Apr 2024

Dear Harmit, 

Thank you for submitting your manuscript entitled "Widespread fungal-bacterial competition for magnesium lowers antibiotic susceptibility" for consideration as a Research Article by PLOS Biology.

We have discussed the revision with an Academic Editor, and while we are in principle interested in the study, we would like to send the manuscript back to the original reviewers to assess the revision.

Once your full submission is complete, your paper will undergo a series of checks in preparation for peer review. After your manuscript has passed the checks it will be sent out for review. To provide the metadata for your submission, please Login to Editorial Manager (https://www.editorialmanager.com/pbiology) within two working days, i.e. by Apr 07 2024 11:59PM.

Kind regards,

Melissa

Melissa Vazquez Hernandez, Ph.D.

Associate Editor

PLOS Biology

---

## [Decision Letter · Decision Letter 1]

8 May 2024

Dear Harmit,

Thank you for your patience while we considered your revised manuscript "Widespread fungal-bacterial competition for magnesium lowers antibiotic susceptibility" for publication as a Research Article at PLOS Biology. This revised version of your manuscript has been evaluated by the PLOS Biology editors, the Academic Editor and the original reviewers. I am so sorry for taking this long. 

Based on the reviews, we are likely to accept this manuscript for publication, provided you satisfactorily address the remaining points raised by the reviewers. Please also make sure to address the following data and other policy-related requests.

a) Please revise figure 5 as suggested by reviewer 3. After discussion with the Academic Editor, we believe that no further experimental work is necessary.

b) Please address our Data Policy requests below; specifically, we need you to supply the numerical values underlying Figs 1DEF, 2ABCDEFG, 3CDFG, 4BCD, S1AB, S2, S3ABC, S4BCD, S5, S7ABC, S8, S9ABCDEFGH, S10, S11AB, S12, S13AB, S14ABCD, S15AB, either as a supplementary data file or as a permanent DOI’d deposition. Please also provide the fold change values for each replicate in S1 Table

c) Please cite the location of the data clearly in all relevant main and supplementary Figure legends, e.g. “The data underlying this Figure can be found in S1 Data” or “The data underlying this Figure can be found in https://doi.org/10.5281/zenodo.XXXXX”

d) Please make any custom code available, either as a supplementary file or as part of your data deposition. You can see our data and code policy at the end of this e-mail

We expect to receive your revised manuscript within two weeks. 

*Published Peer Review History*

*Press*

Sincerely,

Melissa

Melissa Vazquez Hernandez, Ph.D.

Associate Editor

PLOS Biology

DATA POLICY:

1DEF, 2ABCDEFG, 3CDFG, 4BCD, S1AB, S2, S3ABC, S4BCD, S5, S7ABC, S8, S9ABCDEFGH, S10, S11AB, S12, S13AB, S14ABCD, S15AB

CODE POLICY

Per journal policy, if you have generated any custom code during the curse of this investigation, please make it available without restrictions upon publication. Please ensure that the code is sufficiently well documented and reusable, and that your Data Statement in the Editorial Manager submission system accurately describes where your code can be found. [IF APPLICABLE: As the code that you have generated to XXX is important to support the conclusions of your manuscript, its deposition is required for acceptance.]

We require the original, uncropped and minimally adjusted images supporting all blot and gel results reported in an article's figures or Supporting Information files. We will require these files before a manuscript can be accepted so please prepare and upload them now. Please carefully read our guidelines for how to prepare and upload this data: https://journals.plos.org/plosbiology/s/figures#loc-blot-and-gel-reporting-requirements

DATA NOT SHOWN?

REVIEWERS’ COMMENTS

— — — 

Reviewer #1

The authors have satisfactorily answered all the comments from the previous round of review. I have no further concerns.

— — — 

Reviewer #2

This paper looks at the interaction between two co-infecting human pathogens Pseudomonas aeruginosa and Candida albicans. The authors found that C. albicans causes Mg2+ limitation for P. aeruginosa and go on to show the impact of this on P. aeruginosa growth and resistance to the antibiotic coliston. For the most part the experiments are well carried out but in a number places more explanation would seem to be needed:

-Title: "Antibiotic suscepibilty" would seem to be only for polymxyin-like antibiotics, so this should be rewritten to state this. Otherwise it implies that this is true for ALL antibiotics, which does not seem to be the case. Were other antibiotics tested?

-One Sentence Summary and line 339 and line 413: This sounds like the antibiotic is surviving. And this brings up a point, is the activity of colistin dependent on the level the Mg2+ (rather than an effect on Gram negative bacteria)? Has this been tested?

-Abstract and Figure 5 legend: What is non-canonical about this resistance? The low Mg2+ mechanism has been shown previously for P. aeruginosa and Salmonella typhimurium.

-Line 148: Is "~10% lower than WT" really statistically or biologically significant? On the other hand, the increased CFUs of monocultures of mutants compared to WT in Figure 1D are not mentioned.

-Line 391: Is there a reference for the statement about the concentration of Mg2+ during P. aeruginosa infections in humans? Is this all infections or specific ones? If so, why is coliston ever effective?

Figure 5: The designations in this figure are confusing (at least to this reviewer). Why is the symbol "|----" used? Maybe it would be helpful to have a cartoon of what the microbe looks like after coliston treatment. Also, does this imply the mechanisms for resistance to "high" or "low" coliston are different? This should be clarified.

— — — 

Reviewer #3

Generally, my comments are fully addressed.

I recommend additional minor revisions to the model figure in Figure 5A. The changes to Figure 5A for the representation of PmrAB- and PhoPQ-mediated resistance are good. (i) In figure 5A top left, I am confused as to why the Mg2+ and colistin are drawn only in association with the O-antigen portion, it is my understanding the the Mg2+ association with LPS is critical for stabilizing the lipid A and inner core portions which have phosphate groups. For example, Mg2+ stabilization of the envelope is still important for organisms that do not have O-antigen such as Acinetobacter spp. and Neisseria spp. (ii) Similarly, the colistin mechanism of action includes incorporating into the lipid A portion of the outer membrane, not just associating with the O-antigen. (iii) I am also confused why it appears the O-antigen is separating from the lipid A, is this intentional? (iv) Finally, it looks like the authors have represented the outer membrane as comprised primarily of phospholipids with only a couple LPS molecules shown per area, which is not accurate to what we understand about the gram-negative OM; the outer leaflet should be predominantly LPS at least during homeostasis. These are minor issues but I think it is helpful to readers to draw the structures as accurately to what we know as possible.

Overall, this is an exciting, rigorous, and well-written manuscript.

---

## [Editor Report · Decision Letter 2]

29 May 2024

Dear Harmit,

Thank you for the submission of your revised Research Article "Widespread fungal-bacterial competition for magnesium lowers bacterial susceptibility to polymyxin antibiotics" for publication in PLOS Biology. On behalf of my colleagues and the Academic Editor, Aaron Mitchell, I am pleased to say that we can in principle accept your manuscript for publication, provided you address any remaining formatting and reporting issues. These will be detailed in an email you should receive within 2-3 business days from our colleagues in the journal operations team; no action is required from you until then. Please note that we will not be able to formally accept your manuscript and schedule it for publication until you have completed any requested changes.

IMPORTANT: Many thanks for providing the underlying code in GitHub. However, because Github depositions can be readily changed or deleted, please make a permanent DOI’d copy (e.g. in Zenodo) and provide this URL in the manuscript and Data Availability Statement. I have asked my colleagues to include this request alongside their own.

PRESS

Sincerely, 

Melissa

Melissa Vazquez Hernandez, Ph.D., Ph.D.

Associate Editor

PLOS Biology
